# TPR is required for cytoplasmic chromatin fragment formation during senescence

Bethany M Bartlett[1], Yatendra Kumar[1], Shelagh Boyle[1], Tamoghna Chowdhury[2], Andrea Quintanilla[3], Charlene Boumendil[4], Juan Carlos Acosta[3]*, Wendy A Bickmore[1]*

[1]MRC Human Genetics Unit, Institute of Genetics and Cancer, University of Edinburgh, Edinburgh, United Kingdom; [2]Centre for Regenerative Medicine, Institute for Regeneration and Repair, University of Edinburgh, Edinburgh, United Kingdom; [3]Institute of Biomedicine and Biotechnology of Cantabria (CSIC-Universidad de Cantabria), Santander, Spain; [4]Institute of Human Genetics, UMR9002, CNRS – Université de Montpellier, Montpellier, France

*For correspondence:
juan.acosta@unican.es (JCA);
wendy.bickmore@ed.ac.uk
(WAB)

Competing interest: The authors declare that no competing interests exist.

**Abstract** During oncogene-induced senescence there are striking changes in the organisation of heterochromatin in the nucleus. This is accompanied by activation of a pro-inflammatory gene expression programme – the senescence-associated secretory phenotype (SASP) – driven by transcription factors such as NF-κB. The relationship between heterochromatin re-organisation and the SASP has been unclear. Here, we show that TPR, a protein of the nuclear pore complex basket required for heterochromatin re-organisation during senescence, is also required for the very early activation of NF-κB signalling during the stress-response phase of oncogene-induced senescence. This is prior to activation of the SASP and occurs without affecting NF-κB nuclear import. We show that TPR is required for the activation of innate immune signalling at these early stages of senescence and we link this to the formation of heterochromatin-enriched cytoplasmic chromatin fragments thought to bleb off from the nuclear periphery. We show that HMGA1 is also required for cytoplasmic chromatin fragment formation. Together these data suggest that re-organisation of heterochromatin is involved in altered structural integrity of the nuclear periphery during senescence, and that this can lead to activation of cytoplasmic nucleic acid sensing, NF-κB signalling, and activation of the SASP.

## Editor's evaluation

This report provides significant strides in advancing our understanding of how senescence pathway mediated chromatin defects affects genome instability as we age. Their innovative approach, combined with thorough experimental work, provides compelling evidence linking heterochromatin reorganization to the SAHF-CCF-SASP axis. This important work will be of particular interest to the aging, genome instability and cancer fields.

## Introduction

DNA damage, such as short telomeres (replicative senescence) or oncogene signalling, can trigger senescence, an irreversible cell cycle arrest programme. During oncogene-induced senescence (OIS) chromatin organisation is dramatically disrupted. Pre-existing heterochromatin moves away from the nuclear periphery (*Chandra et al., 2012*), forming internal senescence-associated heterochromatic foci (SAHF) (*Narita et al., 2003*).

Senescent cells also activate a gene expression programme that leads to the secretion of a cocktail of inflammatory cytokines, chemokines, and growth factors – known as the senescence-associated secretory phenotype (SASP) (*Coppé et al., 2010*; *Acosta et al., 2013*). The SASP can contribute to tumour suppression by enhancing immune cell recruitment (*Kale et al., 2020*; *Xue et al., 2007*), but it can also promote tumour growth (*Kuilman et al., 2008*) and immunosuppression (*Ruhland et al., 2016*). Activation of SASP-related genes is primarily driven by the transcription factors (TFs) NF-κB (subunit p65) and C/EBPβ (*Chien et al., 2011*; *Kuilman et al., 2008*) and is accompanied by substantial changes in the landscape of active enhancers (*Martínez-Zamudio et al., 2020*; *Tasdemir et al., 2016*).

As well as relocating to the nuclear interior to form SAHF, heterochromatin blebs off from the nuclear membrane during OIS, forming cytoplasmic chromatin fragments (CCFs) (*Ivanov et al., 2013*). The relationship between SAHF, the SASP, and CCFs has been elusive. CCFs are enriched for the heterochromatin-associated histone modifications H3K9me3 and H3K27me3 (*Dou et al., 2017*; *Ivanov et al., 2013*). CCFs are also positive for γ-H2AX, suggesting that DNA damage plays a role in CCF formation (*Ivanov et al., 2013*). In the cytoplasm, CCFs are sensed by the cGAS-STING pathway, which leads to activation of the SASP via NF-κB signalling (*Dou et al., 2017*; *Glück et al., 2017*; *Yang et al., 2017*).

We have previously shown that the nuclear pore basket protein TPR, that excludes heterochromatin from the vicinity of nuclear pores (*Krull et al., 2010*), is necessary for both the formation and maintenance of SAHF, as well as for activation of the SASP, during OIS (*Boumendil et al., 2019*). The AT-hook chromatin protein HMGA1 has similarly been shown to be a component of SAHFs and to be required for SAHF formation (*Narita et al., 2006*). Here, we investigate the requirement of TPR for SASP activation during OIS as well as during the early replicative stress that occurs in response to oncogenic RAS induction. Our results suggest a key role for TPR in the activation of innate immune signalling linked to formation of CCFs. We also show that HMGA1 is required for CCF formation. These data suggest that heterochromatin re-organisation away from the nuclear periphery underlies a loss of nuclear integrity manifesting as CCF formation, and that this can lead to activation of innate immune signalling during senescence.

## Results

### Putative enhancers dependent on TPR during senescence are enriched for binding sites of inflammatory TFs

TPR is a 267 kDa protein (*Figure 1A*) added to the nuclear pore late in telophase after other nuclear pore components and is anchored to the nuclear pore basket through its interaction with NUP153 (*Hase and Cordes, 2003*). Knockdown or degradation of TPR has been shown not to affect NUP153 recruitment to the nuclear pore (*Hase and Cordes, 2003*; *Aksenova et al., 2020*). TPR is necessary for both the formation and maintenance of SAHF, as well as for activation of the SASP, during OIS (*Boumendil et al., 2019*). To further study the role of TPR in the activation of the SASP during OIS, we used IMR90 fibroblasts harbouring an estrogen-inducible (4-hydroxytamoxifen [4-OHT]) oncogenic RAS$^{G12V}$ mutation (ER:HRAS$^{G12V}$) (*Acosta et al., 2013*). The chromatin regulatory landscape of IMR90 cells changes during OIS (*Tasdemir et al., 2016*) and there is evidence that some nucleoporins interact with enhancers and regulate the transcriptional activity of associated genes (*Ibarra et al., 2016*; *Pascual-Garcia et al., 2017*). Therefore, we investigated whether TPR influences putative enhancers that control SASP gene activation. We used ATAC-seq to identify whether there are regions of accessible chromatin that are specific to senescent cells, and that are TPR-dependent – i.e., are lost after TPR depletion by siRNAs at day 8 (d8) of RAS-induced senescence (*Figure 1B*).

Of the 6826 peaks with a significant increase in accessibility in senescent (RAS siCTRL) compared to non-senescent control (STOP siCTRL) cells (senescent-dependent (SEN$^+$)), 1187 are also TPR-dependent (SEN$^+$TPR$^+$) (*Figure 1C*, *Figure 1—figure supplement 1A*, *Supplementary file 1*). Many of these are close to key SASP genes, such as *IL1B* and *IL8* (*Figure 1D*). Both SEN$^+$TPR$^+$ and SEN$^+$TPR$^-$ peak categories showed an increase in H3K27 acetylation (H3K27ac), as assayed from ChIP-seq data (*Parry et al., 2018*), in senescent IMR90 ER:HRAS$^{G12V}$ cells when compared with the non-senescent control (*Figure 1—figure supplement 1B*). This suggests that the regions which become accessible

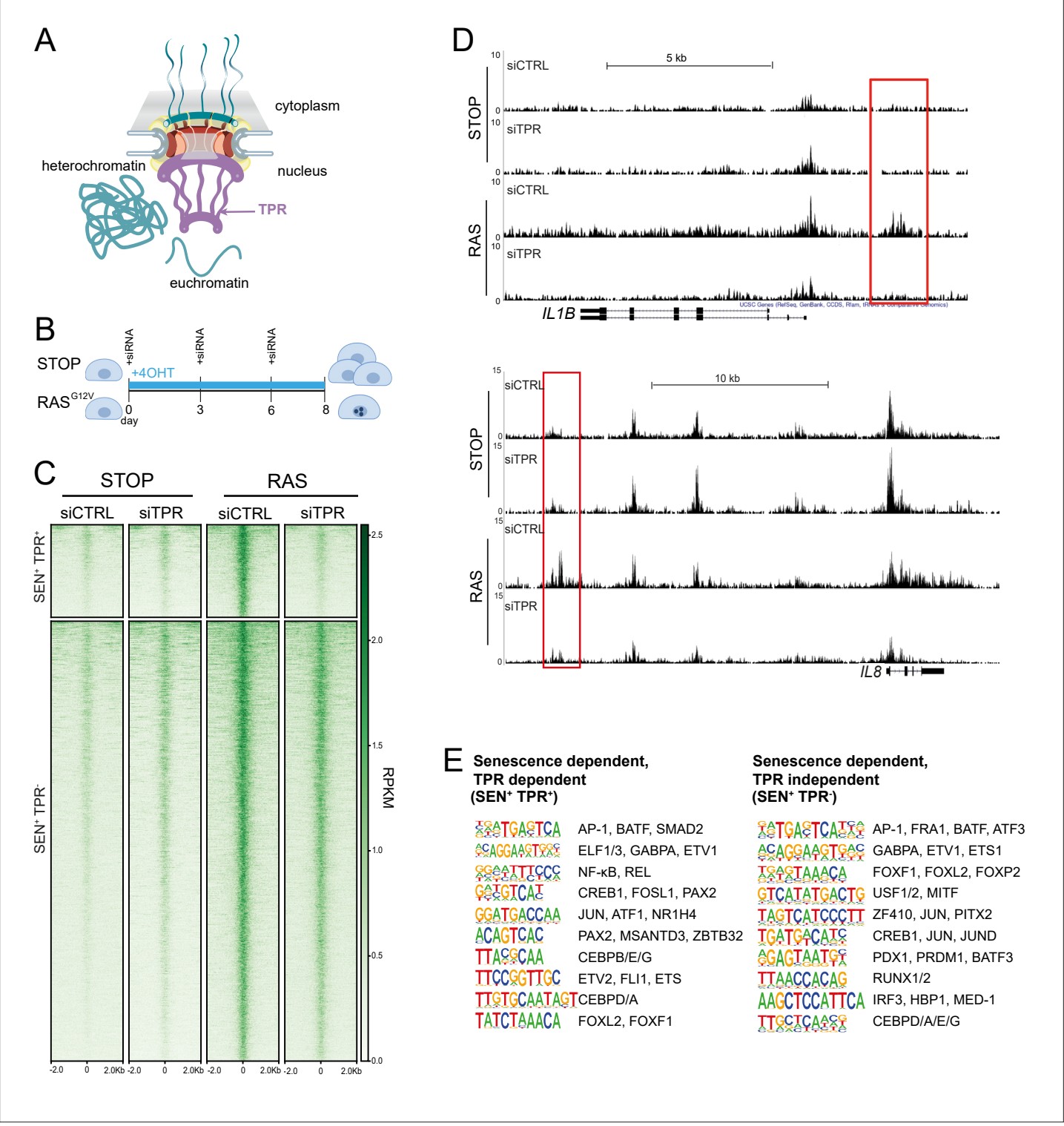

**Figure 1.** Senescence-specific accessible chromatin sites dependent on TPR are near senescence-associated secretory phenotype (SASP) genes and are enriched in binding sites for SASP-related transcription factors. (**A**) Model of the nuclear pore showing the location of TPR in the nuclear basket and heterochromatin exclusion at the pore. (**B**) Schematic of experimental protocol for senescence induction in IMR90 cells. After 8 days of treatment with 4-hydroxytamoxifen (4-OHT), the control (STOP) line continues to proliferate while the RAS line becomes senescent due to induction of RAS$^{G12V}$ expression. (**C**) Heatmap showing ATAC-seq signal in control (STOP) and OIS (RAS) cells 8 days after treatment with 4-OHT and transfection with either control (CTRL) or TPR siRNAs. SEN$^+$ indicates signal specific to senescent cells and TPR$^+$ indicates dependence on TPR. Intensity scale represents reads per kilobase per million mapped reads (RPKM). (**D**) Track views of ATAC-seq data from STOP and RAS cells treated with CTRL or TPR siRNAs at *IL1B*

*Figure 1 continued on next page*

Figure 1 continued

(top) and *IL8* (bottom) gene loci. (**E**) HOMER motif analysis of the senescence and TPR-dependent ATAC-seq peaks (SEN⁺ TPR⁺) and the peaks that are dependent on senescence but not TPR (SEN⁺ TPR⁻). The top 10 motifs are shown for each category of peaks. For both analyses all motifs have a p-value<10⁻¹³.

The online version of this article includes the following figure supplement(s) for figure 1:

**Figure supplement 1.** TPR-dependent senescence-specific accessible chromatin peaks are enriched in H3K27ac and associated with genes relevant to inflammation.

upon senescence may function as senescence-specific enhancers, regardless of their dependence on TPR.

Gene ontology (GO) analysis carried out using the Genomic Regions Enrichment of Annotations Tool (GREAT) (**McLean et al., 2010**) showed that TPR-dependent peaks are significantly near to known SASP factor genes, and to genes enriched in Biological Process and Molecular Function categories such as 'positive regulation of inflammatory response', and genes involved in cytokine activity and cytokine receptor binding (**Figure 1—figure supplement 1C**, **Supplementary file 1**). TPR-independent senescent-dependent peaks showed proximity to chemokine receptor genes (*XCR1*, *CCR1*) (**Figure 1—figure supplement 1D**) whose expression allows cells to sense and respond to chemokines such as those secreted in the SASP (**Coppé et al., 2010**). However, the TPR-independent peaks did not show proximity to any SASP factor genes, suggesting that senescence-activated regulatory elements close to SASP genes (**Tasdemir et al., 2016**) may all be TPR-dependent.

HOMER motif analysis (**Heinz et al., 2010**) revealed that d8 SEN⁺TPR⁺, but not TPR-independent (SEN⁺TPR⁻), ATAC-seq peaks are enriched for binding motifs of TFs, such as NF-κB and C/EBPβ, known to regulate the SASP (**Acosta et al., 2008**; **Kuilman et al., 2008**; **Figure 1E**). This indicates that TPR is involved in regulation of the NF-κB-dependent pro-inflammatory SASP during OIS. Both categories of senescent-dependent peaks are enriched in binding motifs for components of the AP-1 complex (**Figure 1E**), a pioneer TF premarking prospective senescence enhancers (**Martínez-Zamudio et al., 2020**). This suggests that the initial shaping of the senescence regulatory landscape by AP-1 is unaffected by TPR knockdown.

## Prolonged loss of TPR during senescence blocks NF-κB activation

Because of the enrichment for NF-κB motifs in the d8 senescence- and TPR-dependent (SEN⁺TPR⁺) putative enhancers, we set out to investigate whether NF-κB activation is affected by TPR knockdown in senescent cells.

Inactive NF-κB dimers are held in the cytoplasm through their association with IκB proteins. Inducing stimuli trigger activation of the IκB kinase complex (IKK), which leads to phosphorylation and degradation of IκB, allowing the translocation of NF-κB to the nucleus, where it promotes the transcription of target genes (**Hayden and Ghosh, 2012**). We used immunofluorescence to assess NF-κB localisation in the nucleus and in the cytoplasm immediately around the nucleus (see Methods) during OIS and in the presence or absence (siRNA knockdown) of TPR (**Figure 2A**). As expected, NF-κB remained cytoplasmic in control (STOP) cells, but translocation to the nucleus could be detected in senescent RAS cells, with SAHF readily apparent from DAPI staining in the nucleus of these cells. As we previously reported, knockdown of TPR (siTPR) in RAS cells blocks SAHF formation, but it also results in reduced nuclear localisation (decreased nucleocytoplasmic ratio) of NF-κB, consistent with decreased NF-κB activation (**Figure 2A and B**, **Figure 2—figure supplement 1A**, **Figure 2—source data 1**).

Active NF-κB is phosphorylated at serine 536 (**Sakurai et al., 1999**). Immunoblotting showed that, as expected, phosphorylation of NF-κB is increased in RAS cells compared with the STOP control cells. In RAS cells phosphorylation of NF-κB, but not total levels of NF-κB, decreased upon TPR knockdown (**Figure 2C**, **Figure 2—figure supplement 1B**). Phosphorylation of the NF-κB kinase IKK, and total levels of IKKα, were also reduced in RAS cells upon TPR knockdown (**Figure 2D**, **Figure 2—figure supplement 1C**), further suggesting a reduction in NF-κB signalling pathway activation in senescent cells in the absence of TPR.

As TPR is part of the nuclear pore, it is possible that the knockdown of TPR causes a general defect in nuclear import, preventing activated NF-κB being imported into the nucleus upon OIS. To check that this is not the case, we treated control and RAS senescent cells with 4-OHT and siRNAs as before, then exposed them to conditioned media (CM) from either control or RAS cells 8 days post

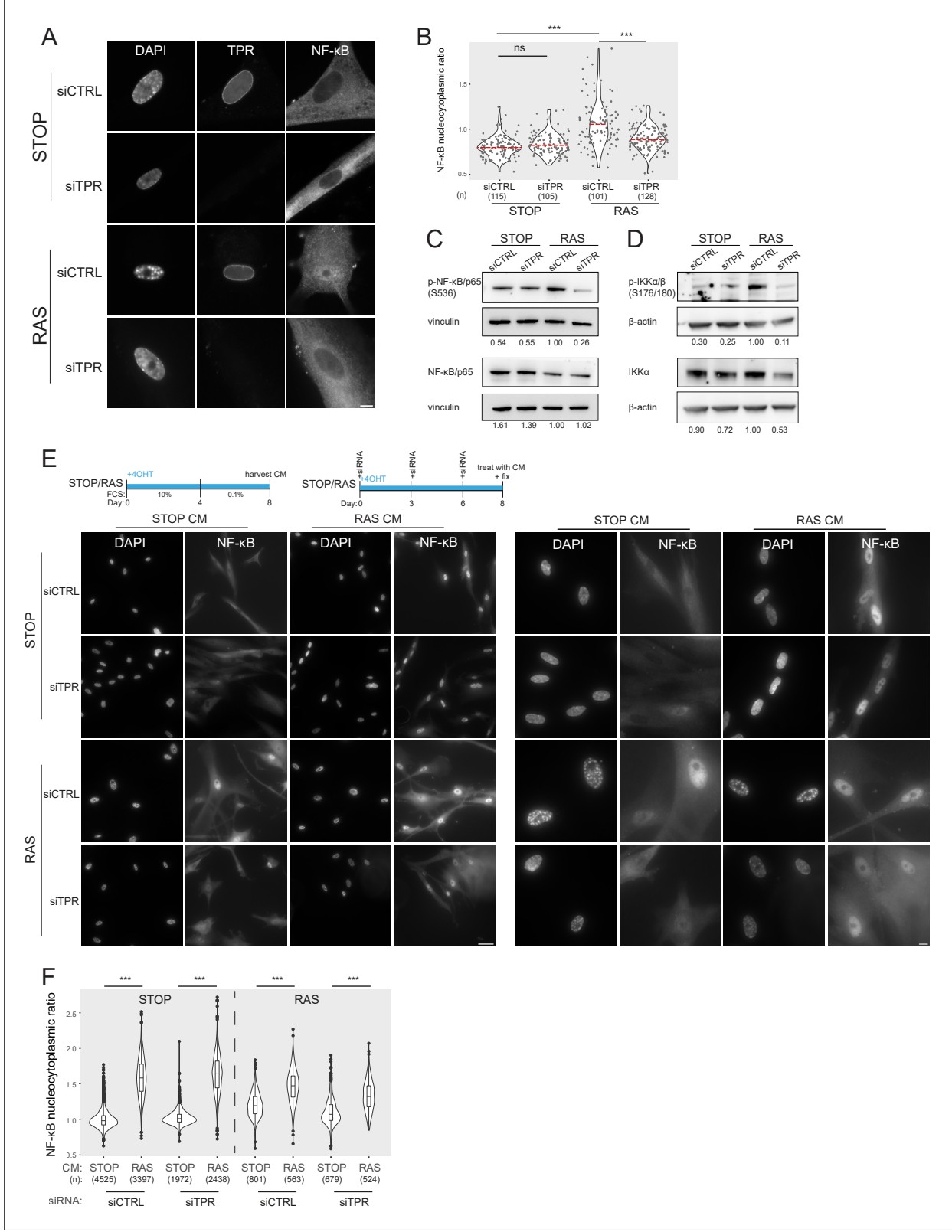

**Figure 2.** Prolonged loss of TPR during senescence blocks NF-$\kappa$B activation. (**A**) TPR and NF-$\kappa$B immunostaining in control (STOP) and oncogene-induced senescence (OIS) (RAS) cells after 4-hydroxytamoxifen (4-OHT) and siRNA (control and TPR) treatment for 8 days. Scale bar: 10 µm. (**B**) Quantification of NF-$\kappa$B nucleocytoplasmic ratios in experiment described in (**A**). Kruskal-Wallis testing was used to determine statistical significance followed by Dunn's post hoc testing. n.s. $p > 0.05$, ***$< 0.001$. (n) indicates the number of cells analysed for each sample. Data from a biological replicate

*Figure 2 continued on next page*

*Figure 2 continued*

are in *Figure 2—figure supplement 1A*. Statistical data are in *Figure 2—source data 1*. (**C**) Immunoblots of extracts from control (STOP) and OIS (RAS) cells after 4-OHT and siRNA treatment for 8 days for phosphorylated (pS536) and total NF-$\kappa$B with vinculin as a loading control. Numbers below indicate the ratio of band intensity for NF-$\kappa$BpS536 or NF-$\kappa$B and the vinculin loading control with the ratio for RAS siCTRL normalised to 1.00. (**D**) As in (**C**) but for phosphorylated (pS176/180) IKK$\alpha$/$\beta$ and total IKK$\alpha$ and with $\beta$-actin as a loading control. Data from biological replicates of (**C**) and (**D**) are in *Figure 2—figure supplement 1B and C*. (**E**) Above: Schematic of controlled media experiment to investigate whether TPR loss causes a general defect in NF-$\kappa$B transport. STOP and RAS cells were grown for 8 days and treated with 4-OHT and siRNAs. On day 8 (d8) they were treated for 45 min with conditioned media (CM) taken from either STOP or RAS cells grown in 4-OHT-containing media for 8 days. Below left: NF-$\kappa$B immunostaining in STOP or RAS cells treated with CM harvested from STOP (left) or RAS (right) cells. Scale bar: 50 µm. Below right: Same experiment with images shown at greater magnification. Scale bar: 10 µm. (**F**) Quantification of NF-$\kappa$B nucleocytoplasmic ratios for experiment shown in (**E**). Data from a biological replicate are in *Figure 2—figure supplement 1D*. Statistical data are in *Figure 2—source data 1*.

The online version of this article includes the following source data and figure supplement(s) for figure 2:

**Source data 1.** Quantification of NF-$\kappa$B nucleocytoplasmic ratios and statistical analysis for data in *Figure 2B and F*, and for biological replicates in *Figure 2—figure supplement 1A and D*.

**Source data 2.** Uncropped and labelled gels for *Figure 2*.

**Source data 3.** Raw unedited gels for *Figure 2*.

**Figure supplement 1.** TPR depletion blocks NF-$\kappa$B activation during senescence.

**Figure supplement 1—source data 1.** Uncropped and labelled gels for *Figure 2—figure supplement 1*.

**Figure supplement 1—source data 2.** Raw unedited gels for *Figure 2—figure supplement 1*.

4-OHT treatment (*Figure 2E*). CM from senescent RAS cells is enriched in SASP factors which leads to NF-κB activation (*Boumendil et al., 2019*). Immunofluorescence showed that nuclear translocation of NF-κB occurs in RAS cells (with control siRNA) in the presence of CM from either STOP or RAS cells, because of their intrinsic activation of the SASP. In STOP cells, nuclear translocation of NF-κB was only induced by CM from RAS cells. This was not affected by TPR knockdown, and this was also the case for RAS cells after TPR knockdown (*Figure 2E*). Quantification of the NF-κB nucleocytoplasmic ratio confirms that TPR knockdown does not affect the nuclear import of NF-κB (*Figure 2F*, *Figure 2—figure supplement 1D*, *Figure 2—source data 1*).

## Decreased NF-κB activation upon TPR knockdown precedes the SASP

The SASP reinforces itself via a positive feedback loop – once secreted, SASP factors bind to receptors on the cell membrane, leading to NF-κB activation and increased SASP (*Figure 3A*; *Acosta et al., 2008*; *Freund et al., 2010*; *Orjalo et al., 2009*). Therefore, the decreased NF-κB activation at d8 of RAS induction upon TPR knockdown could result from a general decrease in the SASP. To determine whether this was the case, we assessed NF-κB nuclear localisation at two earlier timepoints: day 3 (d3), which is before SASP induction and occurs when the cells are coming out of the initial highly proliferative state (*Young et al., 2009*), and day 5 (d5), which is at the initial stages of the inflammatory SASP (*Figure 3B*). There was no change in NF-κB nucleocytoplasmic ratio at d5 between any of the samples, and only a small increase between STOP siCTRL and RAS siCTRL at d3 (*Figure 3C*, *Figure 3—figure supplement 1A*, *Figure 3—source data 1*), suggesting that these timepoints may be too early to observe significant NF-κB nuclear translocation. However, nuclear NF-κB intensity in the cell was increased in OIS-induced RAS cells compared with the control STOP cells at both d3 and d5, suggesting early NF-κB activation (*Figure 3D*, *Figure 3—figure supplement 1B*, *Figure 3—source data 1*). Knockdown of TPR led to significantly lower nuclear NF-κB intensities in RAS cells at both timepoints, suggesting early NF-κB signalling is reduced when OIS is induced in the absence of TPR. A small increase in NF-κB nuclear intensity in d3 STOP cells when TPR was knocked down and a small decrease at d5 were not reproducible (*Figure 3D*, *Figure 3—figure supplement 1B*, *Figure 3—source data 1*). Consistent with an effect on early NF-κB activation, immunoblotting showed that TPR knockdown resulted in decreased NF-κB phosphorylation (S536) in RAS cells at both d3 and d5 (*Figure 3E*, *Figure 3—figure supplement 1C*), and decreased phosphorylation of IKK, the upstream kinase (*Figure 3F*, *Figure 3—figure supplement 1D*). There was no effect of TPR knockdown on total levels of IKKα at these early timepoints.

To determine whether, as at d8, lowered NF-κB activity upon TPR knockdown during early RAS activation (d3) is accompanied by changes in chromatin accessibility at the putative enhancers of

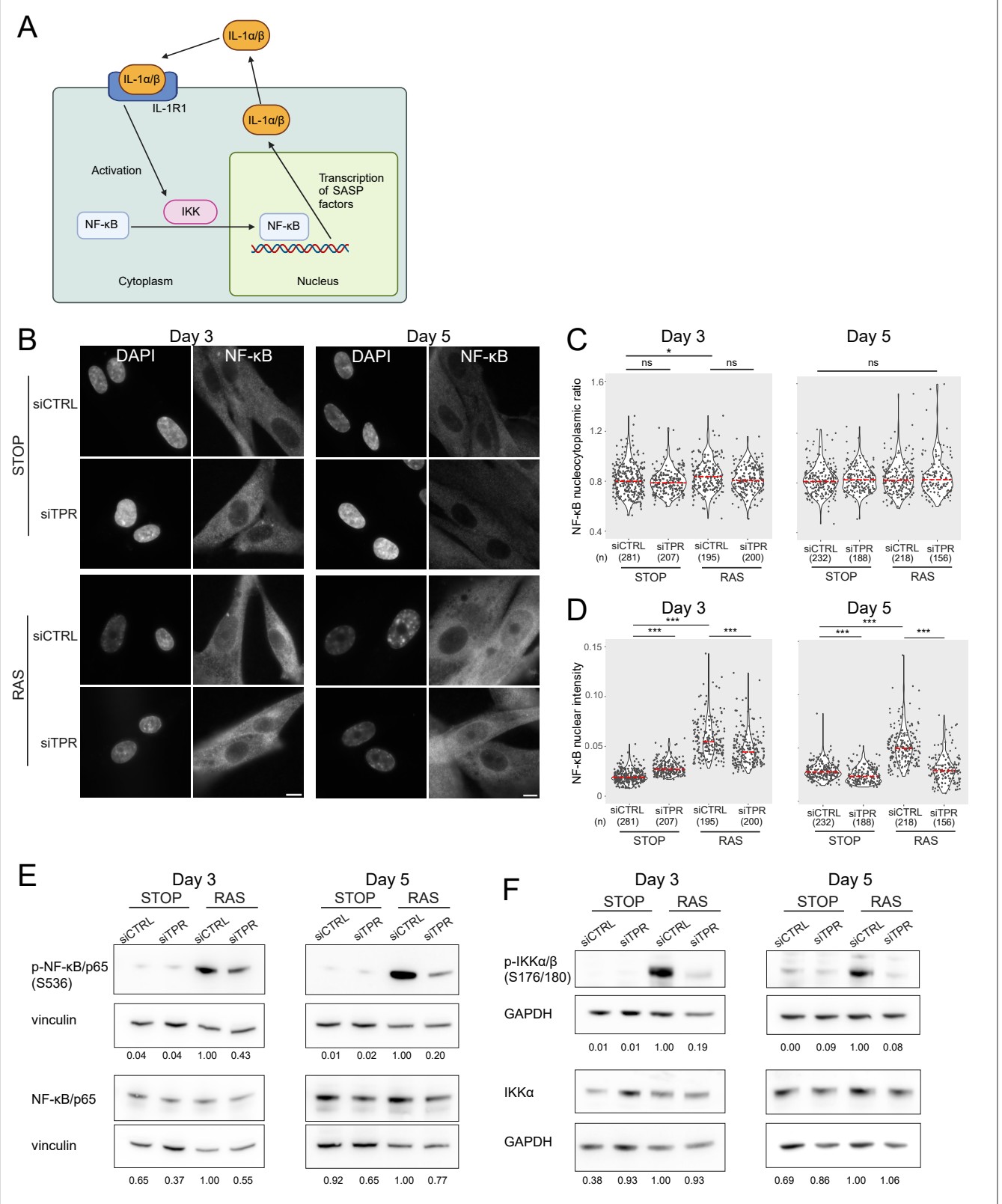

**Figure 3.** Decreased NF-κB activation upon TPR knockdown precedes the senescence-associated secretory phenotype (SASP). (**A**) Schematic showing positive feedback loop in SASP signalling. Secreted IL-1α and IL-1β bind IL-1R1 at the cell membrane, leading to increased NF-κB activation and increased IL-1α and IL-1β secretion. (**B**) NF-κB immunostaining in control (STOP) and oncogene-induced senescence (OIS) (RAS) cells after 4-hydroxytamoxifen (4-OHT) and siRNA treatment for either 3 or 5 days. Scale bar: 10 μm. (**C and D**) Quantification of (**C**) nucleocytoplasmic ratios of

*Figure 3 continued on next page*

*Figure 3 continued*

NF-$\kappa$B or (**D**) NF-$\kappa$B nuclear intensity from experiment shown in (**B**). (**n**) indicates the number of cells analysed for each sample. Kruskal-Wallis testing was used to determine statistical significance followed by Dunn's post hoc testing. n.s. p>0.05, *<0.05, ***<0.001. (**E**) Immunoblots for phosphorylated (pS536) and total NF-$\kappa$B (p65) in STOP and RAS cells treated with 4-OHT for 3 or 5 days and with control (CTRL) or TPR siRNAs. Vinculin was used as a loading control. Numbers below indicate the ratio of band intensity for NF-$\kappa$BpS536 or NF-$\kappa$B and the vinculin loading control with the ratio for RAS siCTRL normalised to 1.00. (**F**) As in (**E**) but blotting to detect phosphorylated (pS176/180) IKK$\alpha/\beta$ and total IKK$\alpha$. GAPDH was used as a loading control. Data from a biological replicate of the data in (**A–E**) are in *Figure 3—figure supplement 1*. Statistical data are in *Figure 3—source data 1*.

The online version of this article includes the following source data and figure supplement(s) for figure 3:

**Source data 1.** Quantification of NF-$\kappa$B nucleocytoplasmic ratios, nuclear intensity, and statistical analysis for data in *Figure 3C and D* and for biological replicates in *Figure 3—figure supplement 1A and B*.

**Source data 2.** Uncropped and labelled gels for *Figure 3*.

**Source data 3.** Raw unedited gels for *Figure 3*.

**Figure supplement 1.** Decreased NF-$\kappa$B activation upon TPR knockdown at days 3 and 5.

**Figure supplement 1—source data 1.** Uncropped and labelled gels for *Figure 2—figure supplement 1*.

**Figure supplement 1—source data 2.** Raw unedited gels for *Figure 2—figure supplement 1*.

**Figure supplement 2.** TPR knockdown does not affect chromatin accessibility at day 3 (d3).

SASP genes, we carried out ATAC-seq on STOP and RAS cells treated with 4-OHT for 3 days, as well as with control and TPR siRNAs. Whilst some of the accessible regions defined as senescence specific (SEN⁺) at d8 also show senescence-specific enhanced chromatin accessibility at d3, albeit less marked than at d8, SEN⁺ accessibility peaks that were TPR-dependent (TPR⁺) at d8 did not show decreased chromatin accessibility upon TPR knockdown at d3 (*Figure 3—figure supplement 2A*). Indeed, we identified no TPR-dependent (TPR⁺) senescence-specific (SEN⁺) ATAC-seq peaks at d3 (*Figure 3—figure supplement 2B and C*, *Supplementary file 1*). Motif analysis showed that the d3 RAS-specific peaks were enriched for AP-1 binding motifs, similar to the d8 TPR⁺SEN⁺ peaks (*Figure 3—figure supplement 2D*). This supports AP-1's role as a pioneer TF in the senescence enhancer landscape (*Martínez-Zamudio et al., 2020*). GO analysis showed that d3 RAS-specific peaks are in proximity to genes involved in SAHF regulation, as well as TGF-β signalling, which has been implicated in the early NOTCH1 regulated SASP (*Figure 3—figure supplement 2E*; *Hoare et al., 2016*).

These data suggest that TPR plays a role in NF-κB activation during the early stages of stress in response to oncogenic RAS, before activation of the SASP and without affecting chromatin accessibility at regulatory elements.

## TPR knockdown during the early stages of OIS reduces STING expression and TBK1 activation in response to the stress induced by oncogenic RAS

Although we detect no changes in chromatin accessibility upon TPR knockdown at d3 of oncogenic stress, the decrease in NF-κB activation suggests that the initial signalling events leading to the loss of the SASP are already occurring. We therefore used RNA sequencing (RNA-seq) to investigate the transcriptional changes that could be driving the TPR-dependent decrease in NF-κB activation at d3.

Through its interaction with the TREX-2 complex, TPR is known to be required for the export of intronless and intron-poor mRNAs, as well as histone mRNAs, the majority of which are intronless (*Aksenova et al., 2020*; *Lee et al., 2020*). Indeed, of the genes downregulated on TPR knockdown, 14% (STOP) or 13% (RAS) are intronless (Fisher's exact test, p=1.2 × 10⁻¹¹; p=7.1 × 10⁻⁹, respectively) (*Figure 4—figure supplement 1A*). This includes histone genes (STOP cells: 3 genes, p=6.1 × 10⁻³; RAS cells: 6 genes, p=4.2 × 10⁻⁶) (*Figure 4—figure supplement 1B*).

*TPR* was the most significantly downregulated gene when comparing siTPR with siCTRL in both RAS and STOP cells (*Figure 4—figure supplement 1C*). To determine which changes in expression were specific to cells undergoing oncogenic stress, we compared RAS siTPR with RAS siCTRL, disregarding any genes that also changed in expression upon TPR knockdown in STOP cells. Interestingly, *STING1* showed the most significant RAS-specific decrease in expression (*Figure 4A*). Reduced *STING1* mRNA in RAS cells after 3 days of RAS induction and TPR knockdown was validated by RT-qPCR (*Figure 4B*, *Figure 4—source data 1*). Immunoblotting did not reproducibly detect reduced levels of STING

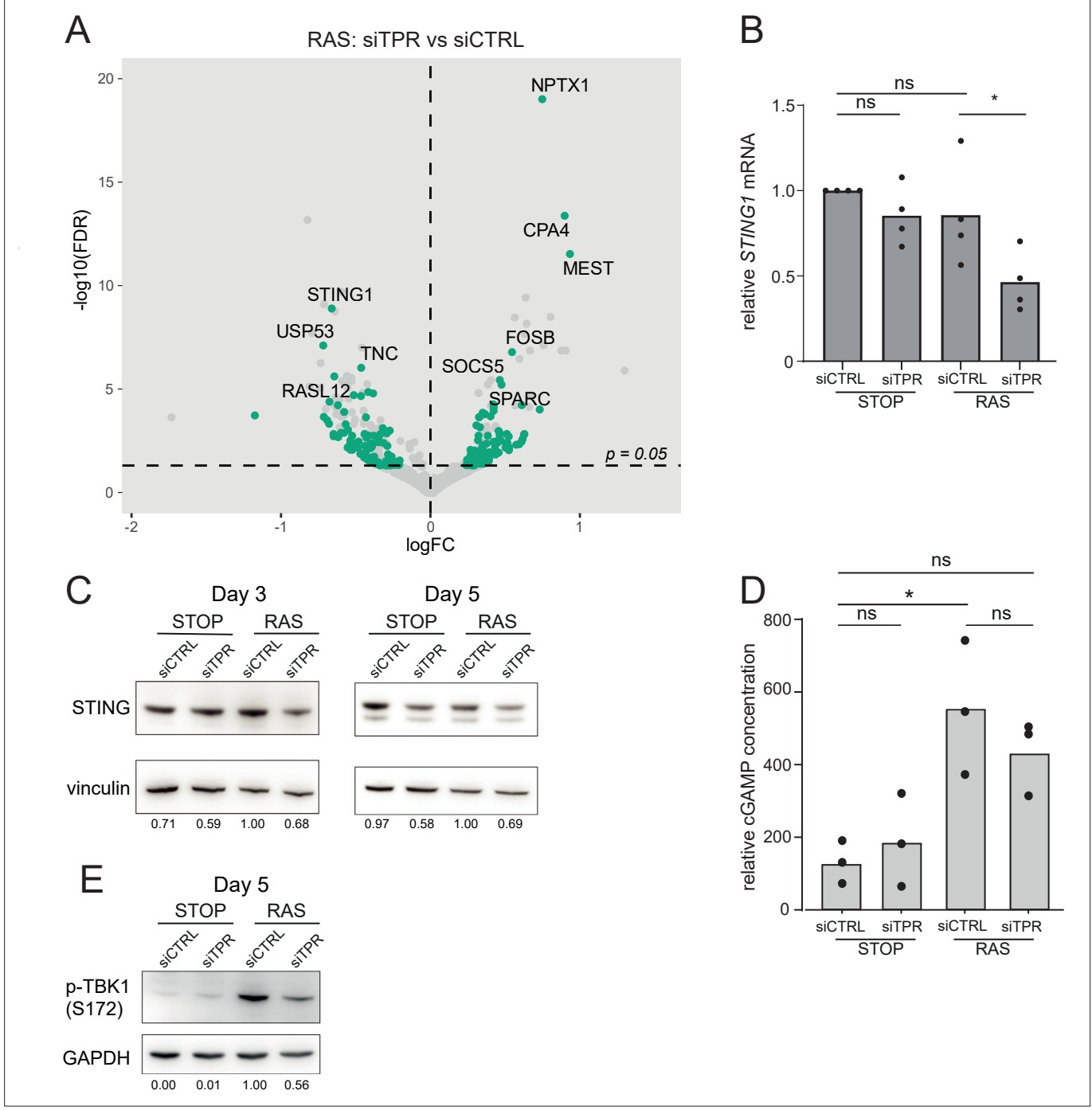

**Figure 4.** Decreased STING expression and TBK1 activation upon TPR knockdown during early stages of oncogene-induced senescence (OIS). (**A**) Volcano plot of differential expression analysis of RNA isolated from RAS cells at day 3 (d3) of OIS and treated with siTPR vs siCTRL. Genes showing a significant change in expression in RAS, but not in STOP cells are indicated in green and the 10 most significant of these are labelled. Horizontal dashed line indicates an adjusted p-value (FDR) of 0.05. Axes are truncated for clarity so the change in *TPR* expression is not shown. (**B**) RT-qPCR for *STING1* mRNA in RNA prepared from STOP and RAS cells treated with 4-hydroxytamoxifen (4-OHT) for 3 days and with control (siCTRL) and TPR siRNAs. Expression is relative to STOP cells treated with siCTRL and normalised to levels of *GAPDH* mRNA. Individual data points are the mean of three technical replicates for each of four biological replicates. Statistical data are in *Figure 4—source data 1*. (**C**) Immunoblots detecting STING in STOP and RAS cells treated with 4-OHT for 3 or 5 days and with control (siCTRL) or TPR siRNAs. Vinculin was used as a loading control. Numbers below indicate the ratio of band intensity for STING and the vinculin loading control with the ratio for RAS siCTRL normalised to 1.00. (**D**) ELISA for 2'3'-cGAMP in STOP and RAS cells treated with 4-OHT for 5 days and with control (siCTRL) or TPR siRNAs. cGAMP concentration was normalised to total protein concentration calculated using BCA assay. Statistical data are in *Figure 4—source data 1*. *p<0.05. (**E**) As in (**C**) but detecting phosphorylated TBK1

*Figure 4 continued on next page*

*Figure 4 continued*

(pS172) in STOP and RAS cells at d5 of OIS. GAPDH was used as a loading control. Data from biological replicates for (**C**) and (**E**) are in *Figure 4—figure supplement 1C and D*.

The online version of this article includes the following source data and figure supplement(s) for figure 4:

**Source data 1.** Statistical analysis for *STING1* qPCR data in *Figure 4B* and for cGAMP ELISA data in *Figure 4D*.

**Source data 2.** Uncropped and labelled gels for *Figure 4*.

**Figure supplement 1.** Decreased abundance of mRNAs for intronless genes and for *STING1* in RAS cells upon TPR knockdown at day 3 (d3).

**Figure supplement 1—source data 1.** Uncropped and labelled gels for *Figure 4—figure supplement 1*.

protein at d3 of RAS induction, perhaps due to protein stability at this short timepoint, but decreased levels were consistently detected by d5 (*Figure 4C*, *Figure 4—figure supplement 1D*).

The cGAS-STING pathway is known to activate the SASP via NF-κB signalling (*Dou et al., 2017*; *Glück et al., 2017*; *Yang et al., 2017*). cGAS-STING detects dsDNA in the cytoplasm, with DNA binding leading to production of 2′3′ cyclic GMP-AMP (cGAMP), a potent STING agonist. However, cGAS-independent STING activation has also been reported (*Unterholzner and Dunphy, 2019*). We assayed the production of cGAMP in STOP and RAS cells and upon TPR knockdown by ELISA (*Figure 4D*). cGAMP was significantly elevated in RAS compared with STOP cells. TPR knockdown in RAS cells appeared to result in decreased cGAMP, though this did not reach statistical significance (*Figure 4—source data 1*).

TANK-binding kinase 1 (TBK1) acts downstream of STING-mediated sensing of cytosolic DNA, and controls NF-κB signalling. TBK1 is phosphorylated at serine 172 when active (*Abe and Barber, 2014*; *Shu et al., 2013*). To investigate whether TPR is required for activation of this pathway early in OIS, we therefore analysed TBK1 phosphorylation. Immunoblotting showed decreased TBK1 phosphorylation in RAS cells upon TPR knockdown at d5 of RAS induction (*Figure 4E*, *Figure 4—figure supplement 1E*). Together, these data are consistent with TPR knockdown blunting STING activation, likely involving cGAS-dependent cytosolic DNA sensing.

The transcription of several classes of retrotransposons, including long-interspersed element-1 (LINE1) and human endogenous retroviruses (HERVs), is known to be activated in senescent cells, and sensed through cGAS triggering an innate immune response (*De Cecco et al., 2019*; *Liu et al., 2023*). Although RNA abundance for some transposable elements, including HERV and LINE1 elements, was higher in RAS compared with STOP cells treated with control siRNA, there were no significant changes in transposable element RNA abundance upon knockdown of TPR in either cell line (*Figure 4—figure supplement 1F*). This suggests that it is not a change in transposable element expression that drives the decrease in innate immune signalling seen upon TPR knockdown at d3 of OIS.

## TPR and HMGA1 are required for the formation of CCFs during the early stages of OIS

Another trigger of innate immune activation in senescent cells is the generation of CCFs (*Dou et al., 2017*; *Glück et al., 2017*; *Yang et al., 2017*). To determine whether TPR affects CCF generation, we assessed their frequency – as evidenced by the proportion of cells with DAPI-stained foci in the cytoplasm not obviously connected to the nucleus - at d3 and d5 of RAS induction. The frequency of detectable CCFs decreased when TPR was knocked down. Though apparent by d3, this was only statistically significant at d5 (*Figure 5A*, *Figure 5—source data 1*).

CCFs form from blebbing off of the nuclear membrane, thought to result from loss of structural integrity of the nuclear envelope (*Ivanov et al., 2013*). CCFs are known to contain lamin B1 (*Dou et al., 2015*) but whether they contain other components of the nuclear envelope is unexplored. By immunostaining we confirmed that, as expected, CCFs are positive for the heterochromatic histone marks H3K9me3 and H3K27me3 (*Figure 5B*) and for γ-H2AX (*Dou et al., 2015*; *Dou et al., 2017*). However, they appear to lack staining for TPR or for POM121, a transmembrane nucleoporin, suggesting that there are no nuclear pores in the CCF envelope (*Figure 5C–E*).

The requirement of TPR for CCF formation during OIS could be a direct consequence of events occurring at the nuclear basket, or could be due to the failure to relocate heterochromatin from the nuclear periphery to internal SAHF when TPR is depleted (*Boumendil et al., 2019*). To distinguish these two scenarios we examined the consequence of HMGA1 knockdown on CCF formation during

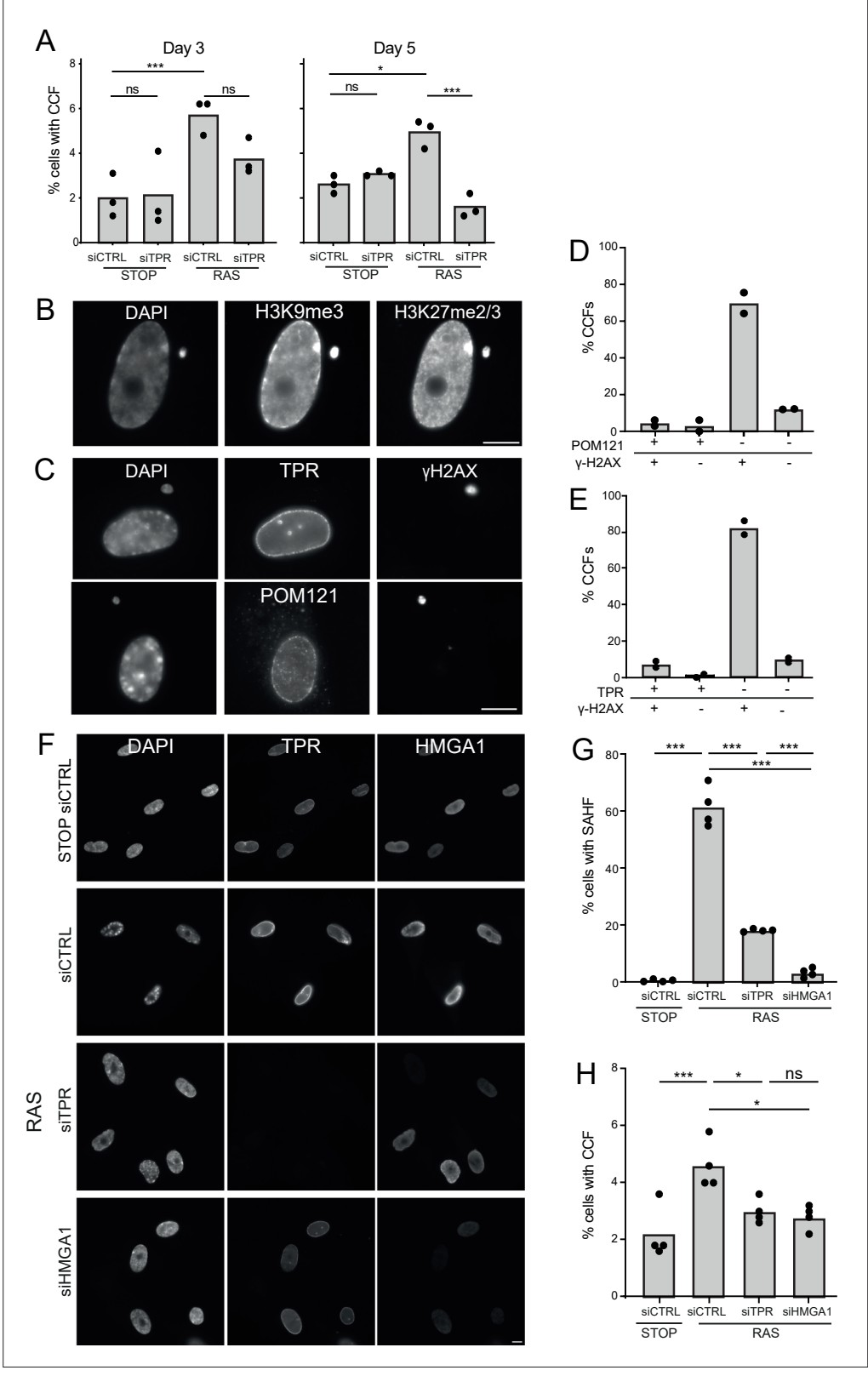

**Figure 5.** TPR and HMGA1 are required for the induction of cytoplasmic chromatin fragments (CCFs) during the early phase of oncogene-induced senescence (OIS). (**A**) Mean percentage of cells containing CCFs in STOP and RAS cells at day 3 (d3) or d5 of OIS and treated with either control (siCTRL) or TPR siRNAs. Data points are for three biological replicates. Data were fitted to a generalised linear model before carrying out pairwise

*Figure 5 continued on next page*

*Figure 5 continued*

comparisons between samples. n.s. p>0.05, *<0.05, ***<0.001. (**B**) Immunostaining for H3K9me3 and H3K27me2/3 in a DAPI-stained d5 RAS cell with a CCF. Scale bar: 10 µm. (**C**) As in (**B**) but in d5 RAS cells containing CCFs and staining for γH2AX and either TPR (top) or POM121 (bottom). Scale bar: 10 µm. (**D**) Mean percentage of CCFs that show +ve or -ve staining for POM121 or γ-H2AX in d5 RAS cells. Data are from two biological replicates (n=49 and 67 CCFs). (**E**) Mean percentage of CCFs that show +ve or -ve staining for TPR or γ-H2AX in d5 RAS cells. Data are from two biological replicates (n=56 and 36 CCFs). (**F**) TPR and HMGA1 immunostaining in control (STOP) and OIS (RAS) cells after 4-hydroxytamoxifen (4-OHT) and siRNA (control, TPR and HMGA1) treatment for 5 days. Scale bar: 10 µm. (**G**) Mean percentage of cells containing senescence-associated heterochromatic foci (SAHF) in STOP and RAS cells at d5 of OIS and treated with either control (siCTRL), TPR, or HMGA1 siRNAs. Data points are for four biological replicates. Data were fitted to a generalised linear model before carrying out pairwise comparisons between samples. *** p<0.001. (**H**) Mean percentage of cells containing CCFs in cells treated as in (**G**). Data points are for four biological replicates. Data were fitted to a generalised linear model before carrying out pairwise comparisons between samples. n.s. p>0.05, *<0.05, ***<0.001. Statistical data from (**A–G**) are in *Figure 5—source data 1*.

The online version of this article includes the following source data for figure 5:

**Source data 1.** Statistical analysis for cytoplasmic chromatin fragments (CCF) and senescence-associated heterochromatic foci (SAHF) data in *Figure 5A, G, and H*.

OIS. HMGA1 is a small AT-rich DNA binding protein abundant in chromatin, that localises to SAHF during OIS. Moreover, depletion of HMGA1 also abrogates SAHF formation (*Narita et al., 2006*; *Olan et al., 2024*). Therefore, we used siRNA to knock down HMGA1 during OIS in RAS cells and examined the effect on CCF formation. Consistent with previous reports, knockdown of HMGA1 in RAS cells led to a significant loss of SAHF compared to control knockdown, and to a greater extent than seen for TPR knockdown (*Figure 5F–G*). HMGA1 depletion was accompanied by a decrease in CCF frequency at d5 similar to that which occurs on TPR knockdown (*Figure 5H*, *Figure 5—source data 1*). These data suggest that the decrease in CCF is linked to changes in heterochromatin organisation, i.e., SAHF formation and not to changes at the nuclear pore basket per se.

## Discussion

We have previously linked TPR at the nuclear basket to the re-organisation of heterochromatin away from the nuclear periphery to form SAHF, and to the activation of SASP genes, during the process of OIS (*Boumendil et al., 2019*). The extent to which these events are coupled was unclear. In this study, we address this by looking at the effects of depleting TPR very early (d3) following the induction of oncogenic RAS, as the cells are responding to the initial stress and before SASP gene transcriptional induction (*Young et al., 2009*).

We show that TPR loss during OIS (d8) prevents chromatin opening at putative SASP gene enhancers enriched in binding motifs for NF-κB – a key TF that drives the SASP. However, we show that TPR is also required for the very early stages of NF-κB activation upon RAS oncogenic stress, well before SASP gene activation, suggesting that TPR does not have a direct effect on chromatin structure at enhancers of the SASP. Rather, our data suggest that TPR loss has its impact upstream of NF-κB and its translocation to the nucleus, by decreasing TBK1 phosphorylation, likely downstream of cGAS-STING signalling. We link this to the production of CCFs – the number of CCFs decreases when TPR is knocked down.

Cytoplasmic chromatin derived from the nuclear genome is known to activate the innate immune response, sensed and signalled through cGAS-STING upstream of TBK1. Cytoplasmic DNA sensing is best studied in the context of micronuclei, formed during mitosis as a consequence of unrepaired DNA damage (*Miller et al., 2021*). Micronuclei can contain many different types of chromatin (*Mammel et al., 2022*) and have been reported to have NPCs in their membrane, albeit at a much lower density than the primary nuclear membrane (*Crasta et al., 2012*; *Hatch et al., 2013*; *Liu et al., 2018*). In contrast, we could not detect a core nuclear pore component (POM121), or TPR, in CCFs, consistent with a mechanism of formation that is distinct from that of micronuclei. This suggests that either the CCFs are formed from the nuclear membrane between NPCs or that NPCs are rapidly lost from CCFs. CCFs form by blebbing off from the nuclear periphery (*Ivanov et al., 2013*; *Miller et al., 2021*) and preferentially contain chromatin fragments enriched in the heterochromatin histone modifications

(H3K9me3 and H3K27me3) that are abundant in lamina-associated domains at the nuclear periphery (*Guelen et al., 2008*). TPR has been suggested to interact with, and affect the organisation of, lamin B1 at the nuclear periphery (*Fišerová et al., 2019*) and there is loss of lamin B1 from the nuclear periphery in senescence (*Dou et al., 2015*; *Freund et al., 2012*; *Sadaie et al., 2013*; *Shimi et al., 2011*). Therefore, it is possible that TPR depletion impacts CCF formation through its effects on lamin B1. However, here we also show decreased CCF formation in OIS RAS cells following knock-down of HMGA1 – a chromatin protein bound throughout the genome (*Olan et al., 2024*) and not a component of the nuclear periphery per se. Like TPR, HMGA1 is also required for heterochromatin re-organisation into SAHF during OIS (*Figure 5*; *Narita et al., 2006*). Therefore our data suggest that CCF formation is linked to the loss of heterochromatin from the nuclear periphery and the formation of SAHF during OIS. Heterochromatin is stiffer and more resistant to deformation than euchromatin (*Ghosh et al., 2021*) and decreasing heterochromatin by inhibiting histone deacetylases has been shown to increase nuclear blebbing (*Stephens et al., 2018*). We therefore consider it likely that the decrease in CCFs produced during the early phases of OIS upon TPR, or HMGA1, knockdown is caused by an increase in the stability of the nuclear periphery due to the heterochromatin that remains there when SAHF are not formed.

Together, our results suggest a role for TPR as an important factor in the loss of nuclear integrity that occurs in response to oncogene-induced stress, leading directly to activation of cytoplasmic nucleic acid sensing and the key inflammatory gene expression programme of senescence. Whether TPR has a similar role for other triggers of senescence and in aging remains to be determined.

## Methods

### Cell culture, CM preparation, and siRNA transfection

IMR90 cells were cultured in DMEM with 10% FBS and 1% penicillin/streptomycin in a 37°C incubator with 5% $CO_2$. IMR90 cells were infected with pLNC-ER:RAS and pLXS-ER:STOP retroviral vectors to produce RAS and STOP cells respectively (*Acosta et al., 2013*). RAS translocation to the nucleus was induced by addition of 100 nM 4-OHT (Sigma). The cell lines tested negative for mycoplasma contamination and their identity is confirmed by their genomic sequence present in ATAC-seq data and their growth response to 4OH. 4-OHT-containing medium was changed every 3 days.

To prepare CM, $5 \times 10^5$ STOP and RAS cells were grown in 100 nM 4-OHT media for 8 days. After 4 days this was replaced with media with 0.1% FCS and 100 nM 4-OHT. Media was harvested on d8. To activate NF-κB, cells were treated with the CM for 45 min.

siRNA knockdown was carried out as previously described (*Boumendil et al., 2019*). Briefly, $9 \times 10^5$ STOP or RAS IMR90 cells (except for imaging experiments, which used $1.5 \times 10^5$ cells) were transfected using Dharmafect transfection reagent (Dharmacon) with a 30 nM final concentration of control (siCTRL, D-001810-10-59), TPR (siTPR, L-010548-00) or HMGA1 (siHMGA1, L-004597-00) siRNA pools (Dharmacon). Transfections were carried out on d0 of 4-OHT treatment and on every third subsequent day.

### Immunofluorescence

Cells were seeded onto coverslips 48 hr before fixation with 4% paraformaldehyde in PBS for 30 min at room temperature, before permeabilisation with 0.2% Triton X-100 for 10 min. Coverslips were then washed three times with PBS before blocking in 1% bovine serum albumin (BSA) for 30 min. Coverslips were then incubated for 45 min in a humid chamber with primary antibody diluted in 1% BSA at the dilutions detailed in the Key resources table. Coverslips were washed three times with PBS. Cells were then incubated with fluorescently labelled secondary antibodies (Life Technologies, Key resources table) for 30 min followed by two washes in PBS. Finally, PBS with 50 ng/ml DAPI was added for 4 min, before a final wash with PBS and mounting onto slides with VectaShield (Vector Laboratories).

Epifluorescence images were acquired using either a Photometrics Coolsnap HQ2 CCD camera (Teledyne Photometrics) or a Hamamatsu Orca Flash 4.0 CMOS camera on a Zeiss Axioplan II fluorescence microscope with Plan-neofluar/apochromat objective lenses (Carl Zeiss UK), a Mercury Halide fluorescent light source (Exfo Excite 120, Excelitas Technologies) and Chroma #83000 triple band pass filter set (Chroma Technology Corp.) with the single excitation and emission filters installed in motorised filter wheels (Prior Scientific Instruments). Image capture was performed using Micromanager

(Version 1.4). For the CM experiment (*Figure 2*) images were acquired using a Photometrics Prime BSI CMOS camera (Teledyne Photometrics) fitted to a Zeiss AxioImager M2 fluorescence microscope with Plan-Apochromat objectives, a Zeiss Colibri 7 LED light source, together with Zeiss filter sets 90 HE, 92 HE, 96 HE, 38 HE, and 43 HE (Carl Zeiss UK). Image capture was performed in Zeiss Zen 3.5 software.

## Image analysis

Nuclear NF-κB intensity and nucleocytoplasmic ratios were calculated using CellProfiler (*Stirling et al., 2021*). Nuclei were identified in the DAPI channel using the Identify Primary Objects module to carry out adaptive Otsu thresholding with a threshold smoothing scale of 5, a threshold correction factor of 0.37, a 200-pixel adaptive window, and a typical object diameter of 100–500 pixels. Clumped objects were distinguished using the 'Intensity' method and dividing lines were drawn between clumped objects using the 'Shape' method. A secondary object was then generated by expanding the primary object by 50 pixels, and NF-κB intensity measured for the primary object (nucleus) and secondary object (nucleus+the cytoplasmic regions immediately surrounding the nucleus). A tertiary object (cytoplasm) was generated by removing the primary object area from the secondary object. Nucleo-cytoplasmic ratio was calculated by dividing the NF-κB intensity in the cytoplasm by the nuclear NF-κB intensity.

To count CCFs, 500 cells per sample were observed by epifluorescence microscopy and cells displaying cytoplasmic DAPI staining were imaged. One blinded replicate was carried out by SB who was unfamiliar with previous results. For quantification of CCFs with γ-H2AX, TPR, and POM121 staining, all cells on a slide of d5 OIS RAS cells were assessed and imaged if they displayed cytoplasmic DAPI staining.

## Immunoblotting

Cells were lysed in Cell Lysis Buffer (20 mM Tris-HCl pH 7.5, 150 mM NaCl, 1 mM $Na_2EDTA$, 1 mM EGTA, 1% Triton X-100, 2.5 mM sodium pyrophosphate, 1 mM β-glycerophosphate, 1 mM $Na_3VO_4$, 1 μg/ml leupeptin, Cell Signaling Technology) with one Pierce Phosphatase and Protease Inhibitor Mini Tablet (Thermo Fisher) added per ml of cell lysate. Protein concentration was quantified using a Pierce BCA protein analysis kit (Thermo Fisher), and then 20 μg of protein was run on NuPage 4–12% Bis-Tris gels (Thermo Fisher) at 150 V for 1 hr. After transfer onto nitrocellulose membranes with an iBlot 2 gel transfer device (Thermo Fisher), membranes were blocked in 5% BSA in TBS with 0.1% Tween-20 (TBS-T) for 30 min then incubated overnight with the primary antibodies at the dilutions detailed in the Key Resources table, in 5% BSA in TBS-T. After 3×10 min washes in TBS-T, membranes were incubated with the appropriate horseradish peroxidase (HRP)-conjugated secondary antibodies, before three further washes with TBS-T. Membranes were imaged using an Amersham ImageQuant 800 imager (Cytiva) on the chemiluminescence setting with the Super-Signal West Femto maximum sensitivity substrate kit (Thermo Fisher). When using the mouse anti-β-actin-HRP antibody, the primary antibody incubation step was omitted and a 10 min incubation carried out alongside the secondary antibody step for other blots, before washing and imaging as before.

Quantification of immunoblots was carried out using Fiji (*Schindelin et al., 2012*). Band intensity was normalised to background intensity and the ratio of the band intensity for the protein of interest divided by the loading control was calculated.

## 2'3'-cGAMP ELISA

2'3'-cGAMP was assayed using the 2'3'-cGAMP ELISA Kit (catalog no. 501700; Cayman Chemical) according to the manufacturer's instructions. Cells were lysed in RIPA buffer (50 mM Tris-HCl pH 7.5, 150 mM NaCl, 0.5% sodium deoxycholate, 0.03% SDS, 0.5 mM $Na_2EDTA$, 0.005% Triton X-100, 1 mM $MgCl_2$, 25 kU benzonase). Each sample was assayed in duplicate. The plate was read at a wavelength of 450 nm. The relative amount of 2'3'-cGAMP was determined by interpolating the intensity values to the standard curve and normalising by total protein concentration, which was determined using a Pierce BCA protein analysis kit (Thermo Fisher).

## ATAC-seq library preparation

A standard ATAC-seq protocol with IMR90 cells yielded too many mitochondrial reads and high PCR duplication levels because of poor tagmentation. To circumvent this issue, we used the Omni-ATAC protocol (*Corces et al., 2017*) with some modifications. Briefly, IMR90 cells were harvested by tryp-sinisation and washed with cold PBS. One million cells were resuspended in ice-cold ATAC resus-pension buffer (ATAC-RSB; 20 mM Tris-HCl pH 7.6, 10 mM $MgCl_2$, 20% dimethyl formamide) and 40 strokes in a 1 ml Dounce using a rounded pestle were applied. Debris was pre-cleared by spinning at 100×$g$ for 3 min. The supernatant was collected and spun again at 1000×$g$ for 5 min to collect the nuclear pellet. The pellet was resuspended in 1 ml ATAC-RSB buffer with 0.1% Tween-20 and spun at 1000×$g$ for 5 min. The nuclear pellet was resuspended in 100 µl TD buffer (10 mM Tris-HCl pH 7.6, 5 mM $MgCl_2$, 10% dimethyl formamide) and the Omni-ATAC protocol performed on $5×10^4$ nuclei. ATAC-seq libraries were made using adaptor sequences as described previously (*Buenrostro et al., 2013*). Libraries were assessed for quality and fragment size using the Agilent Bioanalyzer. Sequencing was performed on the NextSeq 2000 platform (Illumina) using NextSeq 1000/2000 P2 Reagents.

## ATAC-seq data analysis

FastQC was used to obtain basic quality control metrics from sequencing data and to assess the quality of reads before preprocessing steps. Sequencing reads were trimmed to a minimum of 30 bases and adaptor sequences clipped using cutadapt (*Martin, 2011*). Reads were aligned to the human genome assembly hg19 using bowtie2 (*Langmead and Salzberg, 2012*). Mitochondrial reads and PCR dupli-cates were filtered out before shifting reads by +4 bp for the positive strand and –5 bp for the nega-tive strand. Peaks were then called using MACS2 (*Zhang et al., 2008*) before removing all peaks from promoter regions, as we were specifically interested in promoter-distal regulatory elements. The HOMER (*Heinz et al., 2010*) functions makeTagDirectory and annotatePeaks.pl with settings '-noadj -len 0 -size given' were used for read counting before count tables were loaded into RStudio.

Trimmed mean of M-values normalisation was carried out using edgeR (*Robinson et al., 2010*) and analysis of differentially accessible regions was carried out using limma (*Ritchie et al., 2015*). Contrasts were designed as ~0 + Sample, where Sample specifies both the cell line and siRNA treat-ment. A cut-off adjusted p-value of 0.05 was used to define differentially accessible peaks. Heatmaps were generated using the deepTools function plotHeatmap (*Ramírez et al., 2016*). Analysis of nearby genes was carried out using GREAT (*McLean et al., 2010*) with the 'basal plus extension' setting. Motif analysis was carried out using HOMER (*Heinz et al., 2010*).

## Analysis of published ChIP-seq data

NarrowPeak files for H3K27 acetylation ChIP-seq from growing and senescent IMR90 RAS[G12V] cells (*Parry et al., 2018*) were obtained from the Gene Expression Omnibus with accession number GSE103590. Correlation between replicates was checked using the plotCorrelation function from the deepTools package (*Ramírez et al., 2016*). Heatmaps were generated by using the deepTools func-tion plotHeatmap to plot the first replicate from each sample with peak categories taken from the ATAC-seq analysis.

## RT-qPCR

Total RNA was extracted using the RNeasy mini kit (QIAGEN) and cDNAs generated using SuperScript II (Life Technologies). Real-time PCR was performed on a Bio-Rad CFX Touch using SYBR Green PCR master mix (Roche) and primers for STING1 (Fwd; ATATCTGCGGCTGATCCTGC, Rev; TTGTAAGT TCGAATCCGGGC) and GAPDH (Fwd; CAGCCTCAAGATCATCAGCA, Rev; TGTGGTCATGAGTCCT TCCA). Samples were heated at 95°C for 5 min before 44 cycles of 10 s at 95°C, 10 s at 60°C, 20 s at 72°C. Expression was normalised to *GAPDH*.

## RNA-seq library preparation and analysis

Total RNA was extracted using the RNeasy mini kit (QIAGEN). Library preparation was carried out by the Edinburgh Clinical Research Facility from 500 ng of each RNA sample using the NEBNext Ultra II Directional RNA library kit with PolyA enrichment module (New England Biolabs). Libraries were assessed for quality and fragment size using the Agilent Bioanalyzer. Sequencing was performed on the NextSeq 2000 platform (Illumina) using NextSeq 1000/2000 P2 Reagents.

FastQC was used to obtain basic quality control metrics from sequencing data and to assess the quality of reads. For each sample, raw Fastq files were merged and aligned to the genome (hg19) using HISAT2 (*Kim et al., 2019*). Alignment statistics were calculated using GATK (*Van der Auwera and O'Connor, 2020*). Reads were assigned to genomic features using the featureCounts tool from the subread package (*Liao et al., 2014*).

Differential expression analysis between each set of conditions was carried out using DESeq2 (*Love et al., 2014*). Contrasts were carried out between samples, where the sample specifies both the cell line and siRNA treatment. GO analysis was carried out using clusterProfiler (*Wu et al., 2021*). Volcano plots were rendered using ggplot2 (*Wickham, 2016*). A list of intronless genes was obtained from the hg19 GTF file available from UCSC (*Nassar et al., 2023*) by sorting for genes with a single exon. A list of histone genes was obtained from HGNC (*Braschi et al., 2019*).

For the analysis of transposable element expression, raw reads were aligned to the human genome assembly hg38 using HISAT2 (*Kim et al., 2019*). Alignment files were processed using the TEcounts tool from the TEtranscripts pipeline (*Jin et al., 2015*). Resulting transposable element and gene raw counts were then subjected to the variance stabilising transformation in DESeq2 (*Love et al., 2014*) and analysed for differential expression with default settings.

## Statistics

Statistical analysis was performed using R and the specific statistical tests used are described in the relevant text, source data, and figure legends. p-Value significance is denoted on figures as follows: *<0.05, **<0.01, ***<0.001.

## Data availability

RNA-seq and ATAC-seq data generated in this study have been deposited at NCBI GEO GSE264387 and GSE264390, respectively.

## Acknowledgements

We thank the Edinburgh Clinical Research Facility for RNA-seq library preparation and for the sequencing of RNA-seq and ATAC-seq libraries, and the IGC Advanced Imaging facility for their help in fluorescence imaging and image analysis. We are grateful to Marie-Therese El-Daher, IGC, for help with ELISA. Funding Statement: BMB was supported a PhD studentship from the Medical Research Council. YK and WAB were supported by a Wellcome Trust Investigator Award 217120/Z/19/Z. Work in the WAB lab is funded by MRC University Unit grants MC_UU_00007/2 and MC_UU_00035/7. JCA acknowledges funding by Cancer Research UK (CRUK) (C47559/A16243 Training & Career Development Board – Career Development Fellowship), the University of Edinburgh-MRC Chancellor's Fellowship, the Ministry of Science and Innovation of the Government of Spain (Proyecto PID2020-117860GB-I00 financed by MCIN/ AEI /10.13039/501100011033) and the Spanish National Research Council (CSIC). Work in the laboratory of CB is supported by the Centre national de la recherche scientifique (CNRS), the Agence Nationale de la Recherche (ANR), under grant number ANR-21-CE12-0039 (project NPCOS), and the French State within the Plan d'investissements France 2030 (program LabUM EpiGenMed, project ChOICe).

## Additional information

### Funding

| Funder | Grant reference number | Author |
|---|---|---|
| Medical Research Council | MC_UU_00007/2 | Wendy A Bickmore |
| Medical Research Council | MC_UU_00035/7 | Wendy A Bickmore |
| Wellcome Trust | 10.35802/217120 | Yatendra Kumar<br>Wendy A Bickmore |
| Cancer Research UK | C47559/A16243 | Juan Carlos Acosta |

| Funder | Grant reference number | Author |
|---|---|---|
| Ministry of Science and Innovation, Government of Spain | Proyecto PID2020-117860GB-I00 | Juan Carlos Acosta |
| Agence Nationale de la Recherche | ANR-21-CE12-0039 | Charlene Boumendil |

The funders had no role in study design, data collection and interpretation, or the decision to submit the work for publication. For the purpose of Open Access, the authors have applied a CC BY public copyright license to any Author Accepted Manuscript version arising from this submission.

## Author contributions

Bethany M Bartlett, Data curation, Formal analysis, Investigation, Methodology, Writing – original draft, Writing – review and editing; Yatendra Kumar, Data curation, Formal analysis; Shelagh Boyle, Andrea Quintanilla, Investigation; Tamoghna Chowdhury, Formal analysis, Methodology; Charlene Boumendil, Conceptualization, Supervision, Methodology; Juan Carlos Acosta, Conceptualization, Supervision, Funding acquisition, Visualization, Methodology, Writing – original draft, Project administration, Writing – review and editing; Wendy A Bickmore, Conceptualization, Supervision, Funding acquisition, Visualization, Writing – original draft, Project administration, Writing – review and editing

## Author ORCIDs

Bethany M Bartlett ⓘ https://orcid.org/0000-0003-1999-7675
Tamoghna Chowdhury ⓘ https://orcid.org/0009-0004-6287-7227
Charlene Boumendil ⓘ https://orcid.org/0000-0002-1953-3902
Juan Carlos Acosta ⓘ https://orcid.org/0000-0002-7989-7329
Wendy A Bickmore ⓘ https://orcid.org/0000-0001-6660-7735

## Decision letter and Author response

Decision letter https://doi.org/10.7554/eLife.101702.sa1
Author response https://doi.org/10.7554/eLife.101702.sa2

---

# Additional files

### Supplementary files

• Supplementary file 1. *Supplementary file 1a, b, and c* are tables summarising data in the manuscript. (a) Table summarising day 8 ATAC-seq changes in peak accessibility. Number of ATAC-peaks with significant changes generated from comparisons between samples using the limma package with an adjusted p-value cut-off of 0.05. Peaks significantly upregulated in RAS siCTRL compared to STOP siCTRL (SEN+) were further divided into TPR-dependent (SEN+TPR+) and TPR-independent (SEN+TPR-) as shown. (b) Table indicating the proximity of senescence-associated secretory phenotype (SASP) gene promoters to TPR-dependent, senescence-dependent ATAC-seq peaks. Distance (in bp) between TPR+SEN+ ATAC-seq peaks from the transcription start site (TSS) of genes involved in positive regulation of the inflammatory response, cytokine activity, and cytokine receptors. (c) Table summarising day 3 ATAC-seq changes in peak accessibility. Number of peaks with significant changes generated from comparisons between samples using the limma package with an adjusted p-value cut-off of 0.05.

• MDAR checklist

### Data availability

The RNA-seq and ATAC-seq data generated in this study have been deposited in NCBI GEO under Accession numbers GSE264387 and GSE264390, respectively.

The following datasets were generated:

| Author(s) | Year | Dataset title | Dataset URL | Database and Identifier |
|---|---|---|---|---|
| Bickmore WA | 2024 | TPR is required for cytoplasmic chromatin fragment formation during senescence | https://www.ncbi.nlm.nih.gov/geo/query/acc.cgi?acc=GSE264387 | NCBI Gene Expression Omnibus, GSE264387 |
| Bickmore WA | 2024 | TPR is required for cytoplasmic chromatin fragment formation during senescence | https://www.ncbi.nlm.nih.gov/geo/query/acc.cgi?acc=GSE264390 | NCBI Gene Expression Omnibus, GSE264390 |

The following previously published dataset was used:

| Author(s) | Year | Dataset title | Dataset URL | Database and Identifier |
|---|---|---|---|---|
| Parry AJ, Hoare M, Bihary D, Hänsel-Hertsch R | 2018 | H3K27ac ChIP-seq | https://www.ncbi.nlm.nih.gov/geo/query/acc.cgi?acc=GSE103590 | NCBI Gene Expression Omnibus, GSE103590 |

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

# Appendix 1

## Appendix 1—key resources table

| Reagent type (species) or resource | Designation | Source or reference | Identifiers | Additional information |
|---|---|---|---|---|
| Cell line (*Homo sapiens*) | IMR90 STOP cells | *Acosta et al., 2013* | | Generated in the J-C Acosta lab |
| Cell line (*Homo sapiens*) | IMR90 RAS cells | *Acosta et al., 2013* | | Generated in the J-C Acosta lab |
| Antibody | anti-β-actin−HRP (mouse monoclonal) | Sigma-Aldrich | A3854 | WB (1:80000) |
| Antibody | anti-GAPDH (mouse monoclonal) | Abcam | ab125247, RRID:AB 11129118 | WB (1:5000) |
| Antibody | anti-phospho-Histone H2AX (Ser139) (mouse monoclonal) | Merck | 05–636 | IF (1:1000) |
| Antibody | anti-H3K27me2/me3 (mouse monoclonal) | Active Motif | #39536 RRID:AB_2793247 | IF (1:1000) |
| Antibody | anti-H3K9me3 (rabbit polyclonal) | Abcam | ab8898 RRID:AB_306848 | IF (1:2000) |
| Antibody | anti-IKKα (rabbit polyclonal) | Cell Signaling Technology | #2682 RRID:AB_331626 | WB (1:1000) |
| Antibody | anti phospho-IKKα/β (Ser176/180) (rabbit monoclonal) | Cell Signaling Technology | #2697 RRID:AB_2079382 | WB (1:1000) |
| Antibody | anti-NF-κB p65 (mouse monoclonal) | Santa Cruz | sc-8008 RRID:AB_628017 | WB (1:1000), IF (1:100) |
| Antibody | anti-NF-κB p65 (rabbit recombinant monoclonal) | Cell Signaling Technology | #8242 RRID:AB_10859369 | IF (1:500) |
| Antibody | anti-phospho- NF-κB p65 (Ser536) (rabbit recombinant monoclonal) | Cell Signaling Technology | #3033 RRID:AB_331284 | WB (1:500) |
| Antibody | anti-POM121 (rabbit polyclonal) | Genetex | GTX102128 RRID:AB_10732546 | IF (1:500) |
| Antibody | anti-STING (rabbit monoclonal) | Cell Signaling Technology | #13647 RRID:AB_2732796 | WB (1:2000) |
| Antibody | anti-phosphoTBK1 (Ser172) (rabbit monoclonal) | Cell Signaling Technology | #5483 RRID:AB_10693472 | WB (1:1000) |
| Antibody | anti-TPR (rabbit polyclonal) | Abcam | ab84516 | IF (1:500) |
| Antibody | anti-vinculin (rabbit polyclonal) | Abcam | ab91459 RRID:AB_2050446 | WB (1:5000) |
| Antibody | anti-rabbit IgG (H+L) secondary, Alexa Fluor 488 (goat polyclonal) | Invitrogen | A11034 | IF (1:1000) |
| Antibody | anti-mouse IgG (H+L) secondary, Alexa Fluor 568 (donkey polyclonal) | Invitrogen | A10037 | IF (1:1000) |
| Antibody | anti-rabbit IgG, HRP-linked (goat polyclonal) | Cell Signaling Technology | #7074 RRID:AB_2099233 | WB (1:2000) |
| Antibody | anti-mouse IgG, HRP-linked (horse polyclonal) | Cell Signaling Technology | #7076 RRID:AB_330924 | WB (1:2000) |
| Sequence-based reagent | siCTRL | Dharmacon | D-001810-10-59 | ON-TARGETplus siRNA pool |
| Sequence-based reagent | siTPR | Dharmacon | L-010548–00 | ON-TARGETplus siRNA pool |
| Sequence-based reagent | siHMGA1 | Dharmacon | L-004597–00 | ON-TARGETplus siRNA pool |
| Sequence-based reagent | STING1_Fwd | *Dou et al., 2017* | RT-qPCR primer | ATATCTGCGGCTGATCCTGC |
| Sequence-based reagent | STING1_Rev | *Dou et al., 2017* | RT-qPCR primer | TTGTAAGTTCGAATCCGGGC |
| Sequence-based reagent | GAPDH_Fwd | *Dou et al., 2017* | RT-qPCR primer | CAGCCTCAAGATCATCAGCA |
| Sequence-based reagent | GAPDH_Rev | *Dou et al., 2017* | RT-qPCR primer | TGTGGTCATGAGTCCTTCCA |

*Appendix 1 Continued on next page*

*Appendix 1 Continued*

| Reagent type (species) or resource | Designation | Source or reference | Identifiers | Additional information |
|---|---|---|---|---|
| Commercial assay or kit | Pierce BCA protein analysis kit | Thermo Fisher | 23225 | Methods: Immunoblotting |
| Commercial assay or kit | SuperSignal West Femto maximum sensitivity substrate kit | Thermo Fisher | 10095983 | Methods: Immunoblotting |
| Commercial assay or kit | 2'3'-cGAMP ELISA kit | Cayman Chemical | 501700 | Methods: 2'3'-cGAMP ELISA |
| Commercial assay or kit | RNeasy mini kit | Qiagen | 74104 | Methods: RT-qPCR and RNA seq library preparation |
| Commercial assay or kit | NEBNext Ultra II Directional RNA library prep kit | New England Biolabs | E7760 | Methods: RNA seq library preparation |
| Commercial assay or kit | NEBNext Poly(A) mRNA Magnetic Isolation Module | New England Biolabs | E7490 | Methods: RNA seq library preparation |
| Chemical compound, drug | 4-hydroxytamoxifen | Sigma | H7904 | |
| Other | H3K27ac ChIP-seq | *Parry et al., 2018* | NCBI GEO: GSE103590 | See *Figure 1—figure supplement 1* |
| Other | ATAC-seq | This paper | NCBI GEO: GSE264390 | See Methods |
| Other | RNA-seq | This paper | NCBI GEO: GSE264387 | See Methods |
| Software, algorithm | CellProfiler | *Stirling et al., 2021* | RRID:SCR_007358 | |
| Software, algorithm | Micromanager | https://micromanager.org | | Version 1.4 |
| Software, algorithm | FastQC | | RRID:SCR_014583 | |
| Software, algorithm | cutadapt | *Martin, 2011* | RRID:SCR_011841 | |
| Software, algorithm | bowtie2 | *Langmead and Salzberg, 2012* | RRID:SCR_016368 | |
| Software, algorithm | MACS2 | *Zhang et al., 2008*; https://pypi.org/project/MACS2/ | | |
| Software, algorithm | HOMER | *Heinz et al., 2010* | RRID:SCR_010881 | |
| Software, algorithm | edgeR | *Robinson et al., 2010* | RRID:SCR_012802 | |
| Software, algorithm | limma | *Ritchie et al., 2015* | RRID:SCR_010943 | |
| Software, algorithm | deepTools | *Ramírez et al., 2016* | RRID:SCR_016366 | |
| Software, algorithm | GREAT | *McLean et al., 2010* | RRID:SCR_005807 | |
| Software, algorithm | HISAT2 | *Kim et al., 2019* | RRID:SCR_015530 | |
| Software, algorithm | GATK | *Van der Auwera and O'Connor, 2020* | RRID:SCR_015530 | |
| Software, algorithm | subread | *Liao et al., 2014* | RRID:SCR_009803 | |
| Software, algorithm | DeSeq2 | *Love et al., 2014* | RRID:SCR_015687 | |
| Software, algorithm | clusterProfiler | *Wu et al., 2021* | RRID:SCR_016884 | |
| Software, algorithm | ggplot2 | *Wickham, 2016* | RRID:SCR_014601 | |
| Software, algorithm | TEtranscripts | *Jin et al., 2015* | RRID:SCR_023208 | |

