## [Editor Report]

This report provides significant strides in advancing our understanding of how senescence pathway mediated chromatin defects affects genome instability as we age. Their innovative approach, combined with thorough experimental work, provides compelling evidence linking heterochromatin reorganization to the SAHF-CCF-SASP axis. This important work will be of particular interest to the aging, genome instability and cancer fields.

---

## [Decision Letter]

[Editors' note: this paper was reviewed by Review Commons.]

Thank you for submitting your article "TPR is required for cytoplasmic chromatin fragment formation during senescence" for consideration by *eLife*. Your article has been reviewed by 3 peer reviewers at Review Commons, and the evaluation at *eLife* has been overseen by Yamini Dalal as the Senior Editor.

Based on the previous reviews and the revisions, the manuscript has been improved sufficiently for two out of three reviewers. There are some remaining issues that need to be addressed before we can proceed to publication -- one reviewer noted major files were missing in the merged version they reviewed, they ask you to ensure the files are present. The third reviewer is still unconvinced on some key points – I suggest including a rebuttal that specifically addresses those points, and textual clarification that will assist readers of this work.

*Reviewer #1:*

The authors have addressed all of my questions and have conducted additional experiments. However, the manuscript is not yet ready for acceptance. I was unable to locate the tables and supplementary files mentioned, which may have been omitted by mistake. Please ensure these are uploaded. I expect that Table 2, as per my recommendation, is included as claimed in the text. If so, after editorial screening, the manuscript can be accepted. The authors have made significant strides in advancing our understanding of the SAHF-CCF-SASP axis. Their innovative approach, combined with thorough experimental work, provides compelling evidence linking heterochromatin reorganization to CCF formation.

*Reviewer #3:*

In the revised manuscript, the authors claimed that TPR is required for CCF formation during senescence. However, this conclusion is unsupported by the new data. In fact, the results collectively suggest an alternative pathway of how TPR regulates the SASP. As articulated below, there are serious major issues that preclude publication of this manuscript at *eLife*.

1. The study uses a single cell strain IMR90 undergoing a single form of senescence, induced by activated Ras. CCF in this system is less than 6%, arguing against a central role for CCF in mediating the NFkB pathway. Additional systems with more robust CCF must be included.

2. The authors presented new data showing that cGAMP is induced in OIS. However, disruption of TPR had no effect on cGAMP level. This result suggests that cGAS is not affected by TPR, again arguing against the central conclusion that CCF is involved.

3. The results that TPR affects STING expression level suggest that TPR does not go through CCF to regulate NFkB. First, CCF is not known to regulate STING expression level. Second, TPR may affect mRNA processing to regulate STING expression, which again can be independent of CCF. This alternative interpretation makes more sense to this reviewer, as I do not see how CCF is a central player regulated by TPR.

4. The authors suggest that the formation of SAHF is linked to CCF. However, BJ cells undergoing IR and replicative senescence show prominent CCF without SAHF. Again, without testing the generalizability of the finding, the authors overinterpreted the results using a single system.

All in all, the central conclusion of CCF is unsupported by the data. However, I do believe the NFkB and STING data. I suggest that the authors remove CCF data and focus on how TPR regulates STING expression. To establish TPR goes through STING to affect NFkB, I suggest overexpressing STING in TPR knockdown senescent cells, and check if NFkB and SASP can be restored.

---

## [Author Response]

General Statements

We thank the reviewers for taking the time to carefully review our preprint, to highlight the clarity of our experimental data, and to provide their constructive comments. We plan to address these comments as listed below.

Reviewer #1i) "Enhancers dependent on TPR during senescence are enriched for binding sites of inflammatory transcription factors".Proximity to genes does not confirm an enhancer role for that gene, although Tasdemir et al., 2016 suggested this. At that time, HI-C and Hi-CHiP techniques were not well-established. Nowadays, without combining HI-C and H3K27ac ChIP, Hi-ChIP alone cannot definitively identify actual enhancer regions. If we repeatedly use the Tasdemir et al., 2016 map, we risk incorrect mapping of enhancers of SASP. The authors should either use other public Hi-C databases to map the enhancer of SASP or temper their conclusions about enhancers. Otherwise, this could set a precedent for the SASP enhancer region that might not be entirely accurate.The enhancer mapping for SASP is outdated, as advancements in Hi-C have significantly developed this area. Therefore, the claimed enhancers of SASP may not be accurate.

We agree with the reviewer that enhancers are not easy to define, or to pair with their target gene(s). Indeed, we would argue that even combined HI-C and H3K27ac does not define enhancers or enhancer-gene pairs and that the gold-standard evidence for an enhancer is genetics – does its deletion/mutation abrogate gene activation. We would also point out that we did not actually use the Tasdemir data to call enhancers. In response to the reviewer’s comment, we will temper our terminology and now refer to our inter-and intra-genic ATAC-seq peaks only as “putative enhancers”.

ii) “*Many of these include putative enhancers located close to key SASP genes, such as IL1B and IL8 (Figure 1D*).”

I have the same concern as mentioned above (i). However, I am interested in knowing the other key SASP genes where DNA is accessible near the genes. A supplementary table listing key SASP genes along with their distances to the TSS and affected by TPR knock-down would be helpful.

We thank the reviewer for this suggestion. In the revised manuscript, we now provide a table (Table 2) listing the TPR dependent, senescent specific ATAC-seq peaks that are close to genes associated with the ‘positive regulation of inflammatory response’, ‘cytokine activity’ and ‘cytokine receptor binding’ gene ontology terms which were significant in the GREAT analysis, and which includes many SASP genes. This Table also gives the distances (bp) of these regions from the associated gene TSS.

iii) "As we previously reported, knockdown of TPR (siTPR) in RAS cells blocks SAHF formation, but it also results in reduced nuclear localisation (decreased nucleocytoplasmic ratio) of NF-κB, consistent with decreased NF-κB activation (Figure 2A and B, Figure S2A)."TPR is required for CCF, SASP, and SAHF. The relationship between CCF and SASP is well established, but the relationship between SAHF and CCF/SASP remains elusive. Both SAHF and CCF are enriched with heterochromatin markers, suggesting that CCF might originate from SAHF. However, this has not been confirmed. Do the authors think that SAHF is a prerequisite for CCF in the OIS model, or is it an independent event?

We agree with the reviewer that CCFs likely originate from SAHF. Whilst we cannot definitively prove this in our ER-Ras OIS model, in the revised manuscript we intend to further investigate the relationship between SAHF and CCF by knocking down HMGA1 during RAS-induced senescence. Like TPR, HMGA1 depletion is known to lead to loss of SAHF (Narita et al., Cell, 2006) but, unlike TPR, HMGA1 is a chromatin protein enriched on heterochromatin itself. We will assess whether loss of HMGA1 also abrogates CCF formation**.**

iv) The authors suggested that "it is plausible that the decrease in CCFs produced during the early phases of OIS upon TPR knockdown may be caused by an increase in the stability of the nuclear periphery due to the heterochromatin that remains there when SAHF are not formed." I do not completely agree with this explanation because CCF starts forming at day 3-4 but culminates at later time points. According to Figure 5A, only 5-6% of cells are positive for CCFs on day 5. What happens on day 8? By day 8, the percentage of CCF-positive cells could be 20-25%, or the number of CCFs per cell might be 0.2-0.3. If TPR is not required for CCF formation at this stage, then linking CCF to SASP at day 8 becomes critical. This suggests that another mechanism might be driving SASP expression and that TPR could be regulating downstream signaling of CCF. It is possible that changes in nuclear pore density affect the localization of cGAS from the nucleus to the cytoplasm.

In our hands, and using the IMR90 ER-RAS system, CCF formation decreases later in senescence (d8 - only 2% of cells) hence our focus on early timepoints after oncogenic RAS activation. At later timepoints, cGAS activation is also mediated by retrotransposons (de Cecco et al., Nature, 2019; Liu et al., Cell, 2023), as well as leakage of mitochondrial DNA (Victorelli et al., Nature, 2023; Chen et al., Nat. Comms, 2024), and so it is difficult to disentangle the net contribution of these three inputs at the later timepoints**.**

v) Additionally, the authors did not address what happens in the later stages of CCF formation in the absence of TPR. If TPR is not required for CCF formation at later stages, it fails to explain the downstream processes at these time points adequately. This suggests that TPR may also have another mechanism of SASP regulation independent of CCF formation.

In our cellular system CCFs precede the SASP – CCFs are already present at day 3 but SASP factors are not secreted until day 5. However, CCFs are not necessarily required for maintenance of the SASP. Once initiated the SASP is maintained by cytokine feedback loops.

Reviewer #2:1. The claim that TPR knockdown does not affect NFkappaB nuclear translocation indeed stands, but it would be nice if the authors also compared data across conditions in Figure 2F, i.e. siCTRL+Ras CM versus siTPR+Ras CM in RAS cells and provided a p-value as it seems to me that there is some dampening of translocation intensity, which is clearly not the case for STOP cells. The authors focus on this for d3 and d5, but it seems to be also the case for later time points.

As basal NF-κB translocation is lower in RAS cells on TPR knockdown, we would expect a dampening in NF-κB translocation between siCTRL+RAS CM and siTPR+Ras CM regardless of whether there is a transportation defect. In response to the reviewer’s comment we have added a table with median nuclear:cytoplasmic NF-κB ratios and 95% confidence intervals for the data in Fig2B and F and for the NF-κB analyses in Figure 3, and for the corresponding biological replicates in supplementary figures. These tables are now Figure 2 -source data 1 and Figure 3 – source data 1.

2. Also, a comment based on literature or from the authors previous work on TPR, on the extent to which the structural integrity of the nuclear basket is at all affected upon TPR depletion would be helpful for data interpretation.

In the revised manuscript we will refer to the literature showing that TPR is the final component added to the nuclear pore and that its absence does not affect localisation of NUP153 to the nuclear basket (Hase and Cordes., Mol. Biol. Cell 2003; Aksenova et al., Nat Comms, 2020).

3. Magnification of representative cells per each condition in Figure 2E would be welcome.

We have included the requested magnifications in a revised figure 2E.

4. Regarding the data in Figures3 and S3: I am a bit confused about how the obviously decreased NFkappaB nuclear signal (e.g., in Figure 3D) does not translate into a skewed N/C ratio (e.g., in Figure 3C)? The western blots indicate that overall NFkappaB levels remain essentially unchanged? Am I missing something?

As stated in the Methods section, we used a 50-pixel expansion of the detected nuclear area as our cytoplasmic area in the analysis (see Author response image 1). This was because we found detecting and segmenting the whole cytoplasmic area in the NF-κB channel to be unreliable. At day 3 and 5, the decrease in NF-κB nuclear signal in RAS cells on TPR knockdown was accompanied by a decrease in signal in the portion of the cytoplasm closest to the nucleus. This led to no change in the nuclear:cytoplasmic ratio. We believe the redistribution of NF-κB closer to the nucleus in the RAS siCTRL sample indicates early activation and will make this clearer in the revised text. We have also quantified the NF-κB immunoblots (see point 5), to help clarification of this issue**.**

**Author response image 1. sa2fig1:** 

5. Also, along these lines, d8 western blots seem to portray an overall drop in NFkappaB levels. Is this indeed so? Can the authors maybe quantify their blots' replicates and provide a box plot and statistical testing?

We now provide quantification for the NF-κB western blots, indicated below the blots in Figures 2, 3 and 4, and the corresponding supplementary figures, with NF-κB/loading control ratios in RASsiCTRL samples normalised to 1.0. Box plots would not be appropriate as we only have two replicates per experiment**.**

6. Regarding the ATAC-seq data from d3, I think it could be mined a bit more. For example, compare to d8 (which the authors have apparently done, but don't present in detail) and discuss which are these early regions that also become accessible by d3 and what kind of genes and motifs are associated with them. Moreover, the focus in Figure S3E is on ATAC sites shared with d8; how about d3-specific ones? How many of these are there (if any) and how might they be affected?

As shown previously in Table S2, now Table 3 in the revised manuscript, TPR knockdown did not cause any changes in chromatin accessibility at day 3, so there are no day 3 specific TPR dependent peaks. We have made this clearer in the revised text (pg 7). At the reviewer’s suggestion, we have now included motif analysis and GREAT analysis on the day 3 peaks that become accessible in RAS cells but that are not accessible in STOP (RAS-specific peaks). This is shown in a new Figure 3 – figure supplement 2.

7. I trust that the authors quantified their STING blots for the conclusions they present, but since it is difficult to assess these confidently by eye, again, some quantification plots would be welcome in Figures4C,D and S4D,E.

Quantification for the STING western blots is now included in Figure 4 and the corresponding supplementary figure.

8. As controls for Figure 5, it would be interesting to see if active histone readouts also mark CCFs in this system.

Ivanov et al., J. Cell Biol., 2013 showed the absence of H3K9 acetylation from chromatin in CCFs. Further exploration of the types of chromatin/sequences in CCFs is outside the scope of our current manuscript.

9. The POM121 channel in Figure 5C appears to have some small signal foci in the cytoplasm; could these be small CCFs? More generally, the authors focus on these large blobs that only appear in <6% of cells in d3 and d5. Does this increase by d8? What is the effect of TPR knockdown on CCF numbers at that later time point?

The small foci the reviewer is highlighting are background from the POM121 antibody staining rather than CCFs – they do not stain with DAPI, and similar foci are evident in non-senescent cells where CCFs are generally not present. Our unpublished data (see response to Reviewer 1, point iv) from day 8 cells shows that only ~2% of senescent cells are CCF positive regardless of TPR knockdown, which is a similar number to that observed in non-senescent cells at earlier timepoints. Thus, in our hands CCF formation occurs earlier, triggering the SASP, rather than at day 8 when the SASP is already established and reinforced through positive feedback cytokine signalling.

10. I wonder if there is a simple experiment the authors could do to test if this mechanism is only linked to senescence, specifically oncogene-induced senescence? I don't think this is needed to support the conclusions drawn here, but it could significantly broaden the scope of their discovery of, for example, this was true in other senescence models or during proinflammatory activation in general?

These are interesting suggestions, but setting up other senescence models is outside the scope of our current manuscript.

Reviewer #31. The study uses a single cell strain IMR90 undergoing a single form of senescence, induced by activated Ras. To show the generalizability of the finding, the authors are advised to inhibit TPR in other forms of senescence in addition to IMR90. For example, IR or etoposide induces greater amount of CCF than in OIS of IMR90. BJ, MEFs, and ARPE-19 senescence also show prominent CCF.

These are interesting suggestions, but setting up other senescence models is outside the scope of our current manuscript.

2. To convincing show the CCF pathway is involved, the authors need to measure the activity of cGAS-STING pathway. Including cGAMP ELISA will be informative.

We thank the reviewer for this suggestion, and we will try to include this assay in our revised manuscript.

3. The authors used conditioned media to show that TPR KD does not directly affect NFkB nuclear translocation. While this is helpful, conditions other than senescence will be more direct. For example, TNFa treatment or poly I:C transfection induces efficient NFkB nuclear translocation in IMR90 cells.

This experiment (Figure 2EF) was designed to simply show that knocking down TPR does not impair the ability of activated NFkB to enter the nucleus, it is not about senescence per se. Indeed, this is why we included the addition of SASP (RAS) conditioned media to non-senescence STOP cells in Figure 2. We do not think investigating other methods of activating NFkB would add more to the question of whether TPR loss abrogates NFkB nuclear import.

4. Figure 4C and Figure S4D are identical.

Though these STING immunoblots look similar; in fact they are not identical. Below we attach the raw original image in which both biological replicates (Figure 4C and S4D) for Day 3 were run on the same gel as proof of this claim.

5. Figure legend for Figure S4F is mislabeled.

We will correct this.

*Description of the revisions that have already been incorporated in the transferred manuscript*

In response to the reviewer 1’s comment, we now refer to ATAC-seq peaks only as “putative enhancers”.

In response to reviewer 2’s suggestion, in the Introduction we now refer to the literature showing that TPR is the final component added to the nuclear pore and that its absence does not affect localisation of NUP153 to the nuclear basket (Hase and Cordes., Mol. Biol. Cell 2003; Aksenova et al., Nat Comms, 2020).

As suggested by Reviewer 2 we have included magnified images of representative cells per in a revised version of Figure 2E.

The Figure legend for Figure S4F has been corrected.

*Description of analyses that authors prefer not to carry out*

Reviewer 2 suggested that as further controls for Figure 5, it might be interesting to see if active histone readouts also mark CCFs in this system. However, Ivanov et al., J. Cell Biol., 2013 already showed the absence of H3K9 acetylation from chromatin in CCFs. Further exploration of the types of chromatin/sequences in CCFs is outside the scope of our current manuscript.

Reviewers 2 and 3 suggested that it might be interesting – though not essential -to broaden the scope of our discovery of the link between TPR and CCF formation to other senescence models or during proinflammatory activation in general. Setting up other senescence models is outside the scope of our current manuscript and beyond our resources, as the lead author of this manuscript has now finished her PhD and is moving on and a major grant supporting this work has now finished.

Reviewer 3 suggested examining NFkB nuclear translocation after TPR knockdown using other methods of NFkB activation apart from conditioned media from senescent cells. However, the experiment referred to (Figure 2EF) was only designed to show that knocking down TPR does not impair the ability of activated NFkB to enter the nucleus, it is not about senescence per se. We do not think investigating other methods of activating NFkB would add more to the question of whether TPR loss abrogates NFkB nuclear import*.*

[Editors’ note: what follows is the authors’ response to the second round of review.]

Based on the previous reviews and the revisions, the manuscript has been improved sufficiently for two out of three reviewers. There are some remaining issues that need to be addressed before we can proceed to publication -- one reviewer noted major files were missing in the merged version they reviewed, they ask you to ensure the files are present. The third reviewer is still unconvinced on some key points – I suggest including a rebuttal that specifically addresses those points, and textual clarification that will assist readers of this work.Reviewer #1:The authors have addressed all of my questions and have conducted additional experiments. However, the manuscript is not yet ready for acceptance. I was unable to locate the tables and supplementary files mentioned, which may have been omitted by mistake. Please ensure these are uploaded. I expect that Table 2, as per my recommendation, is included as claimed in the text. If so, after editorial screening, the manuscript can be accepted. The authors have made significant strides in advancing our understanding of the SAHF-CCF-SASP axis. Their innovative approach, combined with thorough experimental work, provides compelling evidence linking heterochromatin reorganization to CCF formation.Reviewer #3:In the revised manuscript, the authors claimed that TPR is required for CCF formation during senescence. However, this conclusion is unsupported by the new data. In fact, the results collectively suggest an alternative pathway of how TPR regulates the SASP. As articulated below, there are serious major issues that preclude publication of this manuscript at eLife.1. The study uses a single cell strain IMR90 undergoing a single form of senescence, induced by activated Ras. CCF in this system is less than 6%, arguing against a central role for CCF in mediating the NFkB pathway. Additional systems with more robust CCF must be included.

We appreciate the reviewer’s feedback regarding the use of a single cell strain (IMR90) undergoing oncogene-induced senescence (OIS) via Ras activation. The IMR90 cell line, induced by the ER:RAS system, is a well-established and widely used model to study CCF (cytoplasmic chromatin fragment) formation. It has been validated in previous studies, including Ivanov et al. (Journal of Cell Biology 2013), Dou et al. (Nature 2017), Vizioli et al. (Genes Dev. 2020), Liu et al. (Nature Cell Biology 2021), Zhao et al. (Nature Communications 2020) and Hao et al. (Nature Aging 2024).

In our study, we measured CCFs at early time points (3 & 5 days post Ras induction), which corresponds to the period when we observed effects on NF-κB activation. At this stage, CCFs were detected in less than 10% of cells, consistent with prior findings by Dou et al. Importantly, we chose to investigate these early time points because they coincided with when we observed significant TPR-related effects on NF-κB signaling.

Although the observed CCF frequency (~6%) may seem modest, we argue that this is sufficient to drive NF-κB activation. We propose that paracrine and juxtracrine signaling mechanisms, such as IL1B-mediated effects, could explain how a small fraction of cells influences NF-κB activation and the broader SASP (senescence-associated secretory phenotype) response in the entire cell population. In fact, our experiments on paracrine SASP activation demonstrate that TPR-depleted cells remain responsive to paracrine SASP signaling, leading to NF-κB activation (Figure 2E,F and Figure 2—figure supplement 1D). This further supports the notion that even a seemingly small population of CCF-positive cells can exert a significant influence via paracrine pathways.

Additionally, we acknowledge that our measurements represent a fixed snapshot of the CCF state, without accounting for the dynamic nature of the process, including CCF turnover and degradation. It is plausible that CCFs are part of a highly dynamic process, which could result in underestimation of the actual CCF presence at a given time.

2. The authors presented new data showing that cGAMP is induced in OIS. However, disruption of TPR had no effect on cGAMP level. This result suggests that cGAS is not affected by TPR, again arguing against the central conclusion that CCF is involved.

It is not quite correct to say that TPR has no effect on cGAMP levels. While Ras activation leads to a significant increase in cGAMP levels compared to the control STOP cells in siCTRL sample (Figure 4D and Figure 4 source data 1), this does not occur when TPR is knocked down in RAS vs STOP cells. The reviewer is correct that the decreased cGAMP levels in RAS cells after TPR knockdown, compared with RAS siCTRL, is not statistically significant, and we have clearly stated this in the text.

3. The results that TPR affects STING expression level suggest that TPR does not go through CCF to regulate NFkB. First, CCF is not known to regulate STING expression level. Second, TPR may affect mRNA processing to regulate STING expression, which again can be independent of CCF. This alternative interpretation makes more sense to this reviewer, as I do not see how CCF is a central player regulated by TPR.

We agree with the reviewer on this point. But in our model, TPR contributes to NF-κB regulation not only through its effect on CCF formation but also by influencing STING expression. Together, these effects combine to regulate TBK1 activity and subsequently NF-κB activation and the SASP. To clarify this, we have added a revised model in Figure 5 to better illustrate these interconnected pathways. Further research will be necessary to disentangle the specific contributions of each component of TPR’s effect on SASP regulation and to better understand the dynamics of these processes.

4. The authors suggest that the formation of SAHF is linked to CCF. However, BJ cells undergoing IR and replicative senescence show prominent CCF without SAHF. Again, without testing the generalizability of the finding, the authors overinterpreted the results using a single system.

We agree with the reviewer that we may have over-generalised the link between SAHF formation and CCF generation, as many cells such as BJs can indeed form CCFs without developing SAHFs. We have now clarified this at the end of the Discussion. “Whether TPR has a similar role in other models of OIS, or for other triggers of senescence and in aging where SAHFs may not be present, remains to be determined”.

All in all, the central conclusion of CCF is unsupported by the data. However, I do believe the NFkB and STING data. I suggest that the authors remove CCF data and focus on how TPR regulates STING expression. To establish TPR goes through STING to affect NFkB, I suggest overexpressing STING in TPR knockdown senescent cells, and check if NFkB and SASP can be restored.